# FairCal: Fairness Calibration for Face Verification

**Tiago Salvador**[1,3]**, Stephanie Cairns**[1,3]**, Vikram Voleti**[2,3]**, Noah Marshall**[1,3]**, Adam Oberman**[1,3]
[1] McGill University    [2] Université de Montréal    [3] Mila
[1,3]`{tiago.saldanhasalvador,stephanie.cairns,noah.marshall2}@mail.mcgill.ca`
[2,3] `vikram.voleti@gmail.com`, [1,3] `adam.oberman@mcgill.ca`

## Abstract

Despite being widely used, face recognition models suffer from bias: the probability of a false positive (incorrect face match) strongly depends on sensitive attributes such as the ethnicity of the face. As a result, these models can disproportionately and negatively impact minority groups, particularly when used by law enforcement. The majority of bias reduction methods have several drawbacks: they use an end-to-end retraining approach, may not be feasible due to privacy issues, and often reduce accuracy. An alternative approach is post-processing methods that build fairer decision classifiers using the features of pre-trained models, thus avoiding the cost of retraining. However, they still have drawbacks: they reduce accuracy (AGENDA, PASS, FTC), or require retuning for different false positive rates (FSN). In this work, we introduce the Fairness Calibration (FairCal) method, a post-training approach that simultaneously: (i) increases model **accuracy** (improving the state-of-the-art), (ii) produces **fairly-calibrated** probabilities, (iii) significantly reduces the gap in the **false positive rates**, (iv) does not require knowledge of the **sensitive attribute**, and (v) does not require **retraining**, training an additional model, or retuning. We apply it to the task of Face Verification, and obtain state-of-the-art results with all the above advantages.

## 1 Introduction

Face recognition (FR) systems are being increasingly deployed worldwide in a variety of contexts, from policing and border control to providing security for everyday consumer electronics. According to Garvie et al. (2016), face images of around half of all American adults are searchable in police databases. FR systems have achieved impressive results in maximizing overall **accuracy** (Jain et al., 2016). However, they have also been shown to exhibit significant bias against certain demographic subgroups (Buolamwini & Gebru, 2018; Orcutt, 2016; Alvi et al., 2018), defined by a **sensitive attribute** such as ethnicity, gender, or age. Many FR systems have much higher false positive rates (FPRs) for non-white faces than white faces (Grother et al., 2019). Therefore, when FR is employed by law enforcement, non-white individuals may be more likely to be falsely detained (Allyn, 2020). Thus, it is of utmost importance to devise an easily-implementable solution to mitigate FR bias.

Most efforts to mitigate FR bias have been directed towards learning less biased representations (Liang et al., 2019; Wang et al., 2019b; Kortylewski et al., 2019; Yin et al., 2019; Gong et al., 2020; Wang & Deng, 2020; Huang et al., 2020; Gong et al., 2021). These approaches have enjoyed varying degrees of success: though the bias is reduced, so is the **accuracy** (Gong et al., 2020). They often require **retraining** the models from scratch, which is computationally expensive. Moreover, they often require the **sensitive attribute** of the face (such as ethnicity) (Gong et al., 2020; Dhar et al., 2020; Terhörst et al., 2020a; Dhar et al., 2021; Robinson et al., 2020), which may not be feasible to obtain. Hence, an approach that improves **fairness** in existing models without reducing **accuracy**, or knowing the **sensitive attribute**, or require **retraining**, is missing.

In this work, we focus on the face verification problem, a major subproblem in FR: determine whether two face images depict the same person. Current state-of-the-art (SOTA) classifiers are based on deep neural networks that embed images into a low-dimensional space (Taigman et al., 2014). The "Baseline" method is one that deems a pair of images is a match if the cosine similarity between

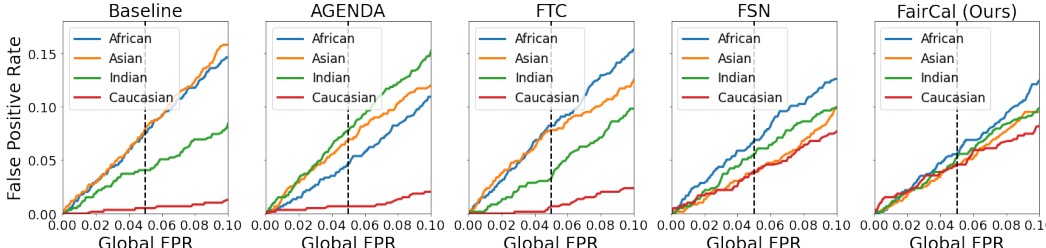

Figure 1: (Lines closer together is better for fairness, best viewed in colour) Illustration of improved fairness / reduction in bias, as measured by the FPRs evaluated on intra-ethnicity pairs on the RFW dataset with the FaceNet (Webface) feature model. The significant subgroup bias in the baseline method is reduced with post-processing methods AGENDA (Dhar et al., 2020), FTC (Terhörst et al., 2020a), FSN (Terhörst et al., 2020b), and FairCal (ours). Our FairCal performs best.

their embeddings exceeds a certain threshold. We illustrate and compare the bias of current models in Figure 1, and compare their characteristics in Table 1.

The fairness of face verification classifiers is typically assessed by three quantities, explained below: (i) **fairness calibration** across subgroups, (ii) equal false positive rates (FPRs) across subgroups i.e. **predictive equality**, (iii) equal false negative rates (FNRs) across subgroups, i.e. equal opportunity.

A classifier is said to be globally calibrated if its predicted probability $p_{pred}$ of face matches in a dataset is equal to the fraction $p_{true}$ of true matches in the dataset (Guo et al., 2017). This requires the classifier to additionally output an estimate of the true probability of a match, i.e. confidence. However, we suggest it is even more crucial for models to be **fairly calibrated**, i.e., for calibration to hold conditionally on each subgroup (see Definition 2). Otherwise, if the same predicted probability is known to carry different meanings for different demographic groups, users including law enforcement officers and judges may be motivated to take **sensitive attributes** into account when making critical decisions about arrests or sentencing (Pleiss et al., 2017). Moreover, previous works such as Canetti et al. (2019) and Pleiss et al. (2017) presume the use of a **fairly-calibrated** model for their success.

While global calibration can be achieved with off-the-shelf post-hoc calibration methods (Platt, 1999; Zadrozny & Elkan, 2001; Niculescu-Mizil & Caruana, 2005; Guo et al., 2017), they do not achieve **fairness-calibration**. This is illustrated in Figure 2. However, achieving all three fairness definitions mentioned above: (i) **fairness-calibration**, (ii) **equal FPRs**, (iii) equal FNRs, is impossible in practice as several authors have shown (Hardt et al., 2016; Chouldechova, 2017; Kleinberg et al., 2017; Pleiss et al., 2017). We elaborate further on this in section 3, the key takeaway is that we may aim to satisfy at most two of the three fairness notions. In the particular context of policing, **equal FPRs** i.e. **predictive equality** is considered more important than equal FNRs, as false positive errors (false arrests) risk causing significant harm, especially to members of subgroups already at disproportionate risk for police scrutiny or violence. As a result, our goals are to achieve **fairness-calibration** and **predictive equality** across subgroups, while maintaining **accuracy**, **without retraining**.

We further assume that we do not have access to the **sensitive attributes**, as this may not be feasible in practice due to (a) privacy concerns, (b) challenges in defining the sensitive attribute (e.g. ethnicity cannot be neatly divided into discrete categories), and (c) laborious and expensive to collect.

In this paper, we succeed in achieving all of the objectives mentioned above. The main contribution is the introduction of a post-training **Fairness Calibration (FairCal)** method:

1. **Accuracy**: FairCal achieves state-of-the-art accuracy (Table 2) for face verification on two large datasets.

2. **Fairness-calibration**: Our face verification classifier outputs state-of-the-art fairness-calibrated probabilities without knowledge of the sensitive attribute (Table 3).

3. **Predictive equality:** Our method reduces the gap in FPRs across sensitive attributes (Table 4). For example, in Figure 2, at a Global FPR of 5% using the baseline method Black people are 15X more likely to false match than white people. Our method reduces this to 1.2X (while SOTA for post-hoc methods is 1.7X).

4. **No sensitive attribute required**: Our approach does not require the sensitive attribute, neither at training nor at test time. In fact, it outperforms models that use this knowledge.

Figure 2: Illustration of bias reduction. False Positives correspond to the imposter pairs above a decision threshold value. *Black horizontal line*: threshold which achieves *global* FPR of 5%; *Red lines*: thresholds for *subgroup* FPRs of 5%. The deviation of the subgroup FPRs from the global FPR is a measure of bias (red lines closer to black horizontal line is better). The baseline method is biased. Calibration (based on cosine similarity alone) does not reduce the bias. The Oracle method reduces bias by using subgroup membership labels for calibration. The FairCal method (ours) reduces bias by using feature vectors for calibration without using subgroup information.

5. **No retraining**: Our method has no need for retraining or tuning the feature representation model, or training an additional model, since it performs statistical calibration *post-hoc*.

## 2 RELATED WORK

**Fairness, or Bias mitigation:** Work on bias mitigation for deep FR models can be divided into two main camps: (i) methods that learn less biased representations, and (ii) post-processing approaches that attempt to remove bias from a pre-trained feature representation model by building fairer decision systems (Srinivas et al., 2019; Dhar et al., 2020; 2021; Terhörst et al., 2020a;b; Robinson et al., 2020). Several approaches have been pursued in (i), including domain adaptation (Wang et al., 2019a), margin-based losses (Khan et al., 2019; Huang et al., 2020), data augmentation (Kortylewski et al., 2019), feature augmentation (Yin et al., 2019; Wang et al., 2019b), reinforcement learning (Wang & Deng, 2020), adversarial learning (Liu et al., 2018; Gong et al., 2020), and attention mechanisms (Gong et al., 2021). While these methods mitigate bias, this is often achieved at the expense of recognition **accuracy** and/or at **high computational cost**. The work we present in this paper fits into (ii), and thus we focus on reviewing those approaches. For an in-depth literature review of bias mitigation for FR models, see Drozdowski et al. (2020). We compare all these methods in Table 1.

Srinivas et al. (2019) proposed an ensemble approach, exploring different strategies for fusing the scores of multiple models. This work does not directly measure bias reduction, and presents only the results of applying the method to a protected subgroup. Terhörst et al. (2020b) show that while this method is effective in increasing overall **accuracy**, it fails to reliably mitigate bias.

Dhar et al. (2020) proposed the Adversarial Gender De-biasing algorithm (AGENDA) to **train** a shallow network that removes the gender information of the embeddings of a pre-trained network. Dhar et al. (2021) extended this work with PASS to deal with any sensitive attribute, and proposed a novel discriminator training strategy. This leads to reduced gender bias in face verification, but at the cost of **accuracy**. Also, AGENDA and PASS require the **sensitive attributes** during training.

Terhörst et al. (2020a) proposed the Fair Template Comparison (FTC) method, which replaces the computation of the cosine similarity score by an **additional** shallow neural network trained using cross-entropy loss, with a fairness penalization term and an $l_2$ penalty term to prevent overfitting. While this method does indeed reduce a model's bias, it comes at the expense of an overall decrease in **accuracy**. Moreover, it requires **training** the shallow neural network and tuning the loss weights. Most importantly, it requires the **sensitive attributes** during training.

Robinson et al. (2020) proposed GST which uses a group-specific threshold. However, the calibration sets in GST are defined by the **sensitive attributes** themselves. Terhörst et al. (2020b) proposed the Fair Score Normalization (FSN) method, which is essentially GST with unsupervised clusters. FSN normalizes the scores by requiring the model's FPRs across unsupervised clusters to be the same at a predefined global FPR. These methods are not designed to be **fairly-calibrated**. Moreover, if a different global FPR is desired, the method needs to be recomputed. In contrast, instead of fixing one FPR threshold, our method converts the cosine similarity scores into calibrated probabilities, which

Table 1: Comparison of desirable features of the different fairness methods for face verification.

| Method (requires re-training) | Improves accuracy | Fairly calibrated | Predictive equality | Does not require sensitive attribute during training | at test time | Does not require re-training |
|---|---|---|---|---|---|---|
| $D^2AE$ (Liu et al., 2018) | ✔ | ✗ | ✗ | ✔ | ✔ | ✗ |
| FTL (Yin et al., 2019) | ✗ | ✗ | ✗ | ✔ | ✔ | ✗ |
| LMFA+TDN (Wang et al., 2019b) | ✗ | ✗ | ✗ | ✗ | ✔ | ✗ |
| SYN (Kortylewski et al., 2019) | ✔ | ✗ | ✗ | ✔ | ✔ | ✗ |
| UMML (Khan et al., 2019) | ✔ | ✗ | ✗ | ✔ | ✔ | ✗ |
| CLMLE (Huang et al., 2020) | ✔ | ✗ | ✗ | ✔ | ✔ | ✗ |
| DebFace-ID (Gong et al., 2020) | ✗ | ✗ | ✔ | ✗ | ✔ | ✗ |
| RL-RBN (Wang & Deng, 2020) | ✔ | ✗ | ✔ | ✗ | ✔ | ✗ |
| GAC (Gong et al., 2021) | ✔ | ✗ | ✔ | ✗ | ✔ | ✗ |

| Method (post-training) | Improves accuracy | Fairly calibrated | Predictive equality | Does not require sensitive attribute during training | at test time | Does not require additional training |
|---|---|---|---|---|---|---|
| AGENDA (Dhar et al., 2020) | ✗ | ✗ | ✔ | ✗ | ✔ | ✗ |
| PASS (Dhar et al., 2021) | ✗ | ✗ | ✔ | ✗ | ✔ | ✗ |
| FTC (Terhörst et al., 2020a) | ✗ | ✗ | ✔ | ✗ | ✔ | ✗ |
| GST (Robinson et al., 2020) | ✔ | ✗ | ✔ | ✗ | ✗ | ✔ |
| FSN (Terhörst et al., 2020b) | ✔ | ✗ | ✔ | ✔ | ✔ | ✔ |
| **Oracle (Ours)** | ✔ | ✔ | ✔ | ✗ | ✗ | ✔ |
| **FairCal (Ours)** | ✔ | ✔ | ✔ | ✔ | ✔ | ✔ |

can then be used for any choice of fair FPRs. In addition, our approach can be extended to group fairness for $K$-classification problems, which is not possible with GST or FSN.

**Calibration:** Calibration is closely related to uncertainty estimation for deep networks (Guo et al., 2017). Several post-hoc calibration methods have been proposed such as Platt's scaling or temperature scaling (Platt, 1999; Guo et al., 2017), histogram binning (Zadrozny & Elkan, 2001), isotonic regression (Niculescu-Mizil & Caruana, 2005), spline calibration (Gupta et al., 2021), and beta calibration (Kull et al., 2017), among others. All of these methods involve computing a calibration function. As such, any calibration method can be readily applied, leading to models that are calibrated but not **fairly-calibrated**. An algorithm to achieve the latter for a binary classifier has been proposed in Hebert-Johnson et al. (2018), but the work remains theoretical and no practical implementation is known. **Fairly calibrated** models are the missing ingredient in the works of Canetti et al. (2019) and Pleiss et al. (2017), since they work with the presumption that a **fairly calibrated** model exists.

## 3 FACE VERIFICATION AND FAIRNESS

In order to discuss bias mitigation strategies and their effectiveness, one must first agree on what constitutes a fair algorithm. We start by rigorously defining the face verification problem as a binary classification problem. Then we present the notion of a probabilistic classifier, which outputs calibrated probabilities that the classification is correct. Finally, since several different definitions of fairness have been proposed (see Verma & Rubin (2018); Garg et al. (2020) for a comparative analysis), we review the ones pertinent to our work.

### 3.1 BINARY CLASSIFIERS AND SCORE OUTPUTS

Let $f$ denote a trained neural network that encodes an image $\boldsymbol{x}$ into an embedding $f(\boldsymbol{x}) = \boldsymbol{z} \in \mathcal{Z} = \mathbb{R}^d$. Pairs of images of faces are drawn from a global pair distribution $(\boldsymbol{x}_1, \boldsymbol{x}_2) \sim \mathcal{P}$. Given such a pair, let $Y(\boldsymbol{x}_1, \boldsymbol{x}_2) = 1$ if the identities of the two images are the same and $Y = 0$ otherwise. The face verification problem consists of learning a binary classifier for $Y$.

The baseline classifier for the face verification problem is based on the cosine similarity between the feature embeddings of the two images, $s(\boldsymbol{x}_1, \boldsymbol{x}_2) = \frac{f(\boldsymbol{x}_1)^T f(\boldsymbol{x}_2)}{\|f(\boldsymbol{x}_1)\|\|f(\boldsymbol{x}_2)\|}$. The cosine similarity score is used to define a binary classifier $\widehat{Y} : \mathcal{X} \times \mathcal{X} \to \{0, 1\}$ by thresholding with a choice of $s_{\text{thr}} \in [-1, 1]$, which is determined by a target FPR: $\widehat{Y}(\boldsymbol{x}_1, \boldsymbol{x}_2) = 1$ if $s(\boldsymbol{x}_1, \boldsymbol{x}_2) \geq s_{\text{thr}}$ and 0 otherwise.

### 3.2 CALIBRATION

A calibrated probabilistic model provides a meaningful output: its confidence reflects the probability of a genuine match, in the sense of the following definition.

**Definition 1.** *The probabilistic model $\widehat{C} : \mathcal{X} \times \mathcal{X} \to [0,1]$ is said to be **calibrated** if the true probability of a match is equal to the model's confidence output $c$: $\mathbb{P}_{\boldsymbol{x}_1, \boldsymbol{x}_2 \sim \mathcal{P}}(Y = 1 \mid \widehat{C} = c) = c$.*

Any score-based classifier can be converted to a calibrated probabilistic model using standard *post-hoc* calibration methods (Zadrozny & Elkan, 2001; Niculescu-Mizil & Caruana, 2005; Platt, 1999; Kull et al., 2017) (see Appendix I for a brief description of ones used in this work). This probabilistic model can be converted to a binary classifier by thresholding the output probability.

Applied to face verification, a calibrated probabilistic model outputs the likelihood/confidence that a pair of images $(\boldsymbol{x}_1, \boldsymbol{x}_2)$ are a match. Calibration is achieved by using a calibration set $S^{\mathrm{cal}}$ to learn a calibration map $\mu$ from the cosine similarity scores to probabilities. Since calibration defines a monotonically increasing calibration map $\mu$, a binary classifier with a score threshold $s_{\mathrm{thr}}$ can be obtained by thresholding the probabilistic classifier at $\mu(s_{\mathrm{thr}})$.

### 3.3 FAIRNESS

In the context of face verification, fairness implies that the classifier has similar performance on different population subgroups, i.e. being calibrated conditional on subgroup membership. Let $g(\boldsymbol{x}) \in G = \{g_i\}$ denote the **sensitive/protected attribute** of $\boldsymbol{x}$, such as ethnicity, gender, or age. Each **sensitive attribute** $g_i \in G$ induces a population subgroup, whose distribution on the intra-subgroup pairs we denote by $\mathcal{G}_i$.

**Definition 2.** *The probabilistic model $\widehat{C}$ is **fairly-calibrated**[1] for subgroups $g_1$ and $g_2$ if the classifier is calibrated when conditioned on each subgroup.*

$$\mathbb{P}_{\boldsymbol{x}_1, \boldsymbol{x}_2 \sim \mathcal{G}_1}(Y = 1 \mid \widehat{C} = c) = \mathbb{P}_{\boldsymbol{x}_1, \boldsymbol{x}_2 \sim \mathcal{G}_2}(Y = 1 \mid \widehat{C} = c) = c.$$

Intuitively, a model is considered biased if its **accuracy** alters when tested on different subgroups. However, in the case of face recognition applications, **predictive equality** (defined below), meaning equal FPRs across subgroups, is often of primary importance.

**Definition 3.** *A binary classifier $\widehat{Y}$ exhibits **predictive equality** for subgroups $g_1$ and $g_2$ if the classifier has equal FPRs for each subgroup,*

$$\mathbb{P}_{(\boldsymbol{x}_1, \boldsymbol{x}_2) \sim \mathcal{G}_1}(\widehat{Y} = 1 \mid Y = 0) = \mathbb{P}_{(\boldsymbol{x}_1, \boldsymbol{x}_2) \sim \mathcal{G}_2}(\widehat{Y} = 1 \mid Y = 0).$$

In certain applications of FR (such as office security, where high FNRs would cause disruption), equal opportunity (defined below), meaning equal FNRs across subgroups, is also important.

**Definition 4.** *A binary classifier $\widehat{Y}$ exhibits **equal opportunity** for subgroups $g_1$ and $g_2$ if the classifier has equal FNRs for each subgroup,*

$$\mathbb{P}_{(\boldsymbol{x}_1, \boldsymbol{x}_2) \sim \mathcal{G}_1}(\widehat{Y} = 0 \mid Y = 1) = \mathbb{P}_{(\boldsymbol{x}_1, \boldsymbol{x}_2) \sim \mathcal{G}_2}(\widehat{Y} = 0 \mid Y = 1).$$

Prior works (Hardt et al., 2016; Chouldechova, 2017; Kleinberg et al., 2017; Pleiss et al., 2017) have shown that it is impossible in practice to satisfy all three fairness properties at the same time: (i) **fairness calibration** (definition 2), (ii) **predictive equality** (definition 3), (iii) equal opportunity (definition 4). At most, two of these three desirable properties can be achieved simultaneously in practice. In the context of our application, **predictive equality** is more critical than equal opportunity. Hence we choose to omit equal opportunity as our goal.

## 4 FAIRNESS CALIBRATION (FAIRCAL)

In this section we describe our FairCal method, which constitutes the main contribution of this work. Simply calibrating a model based on cosine similarity scores will not improve fairness: if the baseline score-based classifier is biased, the resulting probabilistic classifier remains equally biased. This is illustrated in Figure 2, where for both Baseline and Baseline Calibration, any choice of global threshold will lead to different FPRs across **sensitive** subgroups; consequently, these models will fail to achieve **predictive equality**.

---

[1]This notion is also referred to as well-calibration (Verma & Rubin, 2018) and calibration within subgroups (Kleinberg et al., 2017; Berk et al., 2021)

Our proposed solution is to introduce *conditional* calibration methods, which involve partitioning the pairs into sets, and calibrating each set. Given an image pair, we first identify its set membership, and then apply the corresponding calibration map. In our FairCal method, the sets are formed by clustering the images' feature vectors in an unsupervised fashion. In our Oracle method, the sets are defined by the **sensitive attributes** themselves. Both methods are designed to achieve **fairness-calibration**.

### 4.1 FAIRNESS CALIBRATION (FAIRCAL, OUR METHOD)

(i) Apply the $K$-means algorithm to the image features, $\mathcal{Z}^{\text{cal}}$, partitioning the embedding space $\mathcal{Z}$ into $K$ clusters $\mathcal{Z}_1, \ldots, \mathcal{Z}_k$. These form the $K$ calibration sets:

$$S_k^{\text{cal}} = \{s(\boldsymbol{x}_1, \boldsymbol{x}_2) : f(\boldsymbol{x}_1) \in \mathcal{Z}_k \text{ or } f(\boldsymbol{x}_2) \in \mathcal{Z}_k\}, \quad k = 1, \ldots, K \tag{1}$$

(ii) For each calibration set $S_k^{\text{cal}}$, use a post-hoc calibration method to compute the calibration map $\mu_k$ that maps scores $s(\boldsymbol{x}_1, \boldsymbol{x}_2)$ to cluster-conditional probabilities $\mu_k(s(\boldsymbol{x}_1, \boldsymbol{x}_2))$. We use beta calibration (Kull et al., 2017), the recommended method when dealing with bounded scores.

(iii) For an image pair $(\boldsymbol{x}_1, \boldsymbol{x}_2)$, find the clusters they belong to. If both images fall into the same cluster $k$, define the pair's calibrated score as the cluster's calibrated score:

$$c(\boldsymbol{x}_1, \boldsymbol{x}_2) = \mu_k(s(\boldsymbol{x}_1, \boldsymbol{x}_2)) \tag{2}$$

Else, if the images are in different clusters, $k_1$ and $k_2$, respectively, define the pair's calibrated score as the weighted average of the calibrated scores in each cluster:

$$c(\boldsymbol{x}_1, \boldsymbol{x}_2) = \theta \, \mu_{k_1}(s(\boldsymbol{x}_1, \boldsymbol{x}_2)) \; + \; (1 - \theta) \, \mu_{k_2}(s(\boldsymbol{x}_1, \boldsymbol{x}_2)), \tag{3}$$

where the weight $\theta$ is the relative population fraction of the two clusters,

$$\theta = \left|S_{k^1}^{\text{cal}}\right| / (\left|S_{k^1}^{\text{cal}}\right| + \left|S_{k^2}^{\text{cal}}\right|) \tag{4}$$

Since FairCal uses unsupervised clusters, it does not require knowledge of the **sensitive attributes**. Moreover, FairCal is a post-training statistical method, hence it does not require any **retraining**.

### 4.2 COMPARISON WITH FSN (TERHÖRST ET AL., 2020B)

Building the unsupervised clusters allows us to not rely on knowing the **sensitive attributes**. In contrast, the FSN method normalizes the scores by shifting them such that thresholding at a predefined global FPR leads to that same FPR on each calibration set. This is limiting in two crucial ways: 1) FSN only applies to the binary classification problem (e.g. face verification); 2) the normalizing shift in FSN depends on the global FPR chosen a priori. Our FairCal method has neither of these limitations. By converting the scores to calibrated probabilities, we can extended it to the multi-class setting, and we do not need to choose a global FPR a priori.

A simple analogy to explain the difference is : consider the problem of fairly assessing two classrooms of students with different distribution of grades. Fair calibration means the same grade should mean the same for both classrooms. Equal fail rate implies the percentage of students who fail is the same in both classrooms. Global (unfair) methods would choose to pass or fail students from both classrooms using the same threshold. Two possible fair approaches are: (A) Estimate a different threshold for each classroom based on a fixed fail rate for both. (B) Calibrate the grades so the distributions of the two classrooms match, i.e. the same grade means the same in both classrooms, and then choose a fair threshold. Method B automatically ensures equal fail rate for both classes. Method A is what FSN does, Method B is what our FairCal method achieves.

### 4.3 ORACLE CALIBRATION (SUPERVISED FAIRCAL, ALSO OURS)

We include a second calibration method, Oracle, which proceeds like the FairCal defined above, but creates the calibration sets based on the **sensitive attribute** instead in an unsupervised fashion. If the images belong to different subgroups i.e. $g(\boldsymbol{x}_1) \neq g(\boldsymbol{x}_2)$, then the classifier correctly outputs zero. This method is not feasible in practice, since the **sensitive attribute** may not be available, or because using the sensitive attribute might not permitted for reasons of discrimination or privacy. However, the Oracle method represents an ideal baseline for **fairness-calibration**.

## 5 EXPERIMENTAL DETAILS

**Models**: We used three distinct pretrained models: two Inception Resnet models (Szegedy et al., 2017) obtained from Esler (2021) (MIT License), one trained on the VGGFace2 dataset (Cao et al., 2018) and another on the CASIA-Webface dataset (Yi et al., 2014), and an ArcFace model obtained from Sharma (2021) (Apache 2.0 license) and trained on the refined version of MS-Celeb-1M (Guo et al., 2016). We will refer to the models as Facenet (VGGFace2), Facenet (Webface), and ArcFace, respectively. As is standard for face recognition models, we pre-processed the images by cropping them using a Multi-Task Convolution Neural Network (MTCNN) algorithm (Zhang et al., 2016). If the algorithm failed to identify a face, the pair was removed from the analysis.

**Datasets**: We present experiments on two different datasets: Racial Faces in the Wild (RFW) (Wang et al., 2019a) and Balanced Faces in the Wild (BFW) (Robinson et al., 2020), both of which are available under licenses for non-commercial research purposes only. Both datasets already include predefined pairs separated into five folds. The results we present are the product of leave-one-out cross-validation. The RFW dataset contains a 1:1 ratio of genuine/imposter pairs and 23,541 pairs in total (after applying the MTCNN). The dataset's images are labeled by ethnicity (African, Asian, Caucasian, or Indian), with all pairs consisting of same-ethnicity images. The BFW dataset, which possesses a 1:3 ratio of genuine/imposter pairs, is comprised of 890,347 pairs (after applying the MTCNN). Its images are labeled by ethnicity (African, Asian, Caucasian, or Indian) and gender (Female or Male), and it includes mixed-gender and mixed-ethnicity pairs. The RFW and BFW datasets are made up of images taken from MS-Celeb-1M (Guo et al., 2016) and VGGFace (Cao et al., 2018), respectively. Thus, to expose the models to new images, only the two FaceNet models can be evaluated on the RFW dataset, while only the FaceNet (Webface) and ArcFace models can be evaluated on the BFW dataset.

**Methods**: For both the FSN and FairCal method we used $K = 100$ clusters for the $K$-means algorithm, as recommended by Terhörst et al. (2020b). For FairCal, we employed the recently proposed beta calibration method (Kull et al., 2017). Our FairCal method is robust to the choice of number of clusters and post-hoc calibration method (see Appendix J for more details). We discuss the parameters used in training AGENDA (Dhar et al., 2020), PASS (Dhar et al., 2021) and FTC (Terhörst et al., 2020a) in Appendix B, in Appendix C and Appendix D, respectively.

Notice that the prior relevant methods (Baseline, AGENDA, PASS, FTC, GST, and FSN) output scores that, even when rescaled to [0,1], do not result in calibrated probabilities and hence, by design, *cannot* be **fairly-calibrated**. Therefore, in order to fully demonstrate that our method is superior to those approaches, when measuring **fairness-calibration** we apply beta calibration to their final score outputs as well, which is the same post-hoc calibration method used in our FairCal method.

## 6 RESULTS

In this section we report the performance of our models with respect to the three metrics: (i) **accuracy**, (ii) **fairness calibration**, and (iii) **predictive equality**. Our results show that among post hoc calibration methods,

1. FairCal is best at global **accuracy**, see Table 2
2. FairCal is best on **fairness calibration**, see Table 3.
3. FairCal is best on **predictive equality**, i.e., equal FPRs, see Table 4.
4. FairCal **does not require the sensitive attribute**, and outperforms methods that use this knowledge, including a variant of FairCal that uses the sensitive attribute (Oracle).
5. FairCal **does not require retraining** of the classifier.

We discuss these results in further detail below. We provide additional detailed discussion and results, including on equal opportunity and the standard deviations that result from the 5-fold cross-validation study, in the Appendix. Overall, our method satisfies both fairness definitions without decreasing baseline model **accuracy**. In contrast, while the FTC method obtains slightly better **predictive equality** results than our method in one situation, this is achieved only at the expense of a significant decrease in **accuracy**.

Table 2: Global **accuracy** (higher is better) measured by AUROC, and TPR at two FPR thresholds.

| | RFW | | | | | | BFW | | | | | |
| | FaceNet (VGGFace2) | | | FaceNet (Webface) | | | FaceNet (Webface) | | | ArcFace | | |
| (↑) | AUROC | TPR @ 0.1% FPR | TPR @ 1% FPR | AUROC | TPR @ 0.1% FPR | TPR @ 1% FPR | AUROC | TPR @ 0.1% FPR | TPR @ 1% FPR | AUROC | TPR @ 0.1% FPR | TPR @ 1% FPR |
|---|---|---|---|---|---|---|---|---|---|---|---|---|
| Baseline | 88.26 | 18.42 | 34.88 | 83.95 | 11.18 | 26.04 | 96.06 | 33.61 | 58.87 | 97.41 | 86.27 | 90.11 |
| AGENDA | 76.83 | 8.32 | 18.01 | 74.51 | 6.38 | 14.98 | 82.42 | 15.95 | 32.51 | 95.09 | 69.61 | 79.67 |
| PASS | 86.96 | 13.67 | 29.30 | 81.44 | 7.34 | 20.93 | 92.27 | 17.21 | 38.32 | 96.55 | 77.38 | 85.26 |
| FTC | 86.46 | 6.86 | 23.66 | 81.61 | 4.65 | 18.40 | 93.30 | 13.60 | 43.09 | 96.41 | 82.09 | 88.24 |
| GST | 89.57 | 22.61 | 40.72 | 84.88 | 17.34 | 31.56 | 96.59 | 44.49 | 66.71 | 96.89 | 86.13 | 89.70 |
| FSN | 90.05 | 23.01 | 40.21 | 85.84 | 17.33 | 32.80 | 96.77 | **47.11** | 68.92 | 97.35 | 86.19 | 90.06 |
| **FairCal (Ours)** | **90.58** | **23.55** | **41.88** | **86.71** | **20.64** | **33.13** | **96.90** | 46.74 | **69.21** | **97.44** | **86.28** | **90.14** |
| *Oracle (Ours)* | *89.74* | *21.40* | *41.83* | *85.23* | *16.71* | *31.60* | *97.28* | *45.13* | *67.56* | *98.91* | *86.41* | *90.40* |

Table 3: **Fairness calibration** measured by the mean KS across the sensitive subgroups. **Bias:** measured by the deviations of KS across subgroups: Average Absolute Deviation (AAD), Maximum Absolute Deviation (MAD), and Standard Deviation (STD). (Lower is better in all cases.)

| | RFW | | | | | | | | BFW | | | | | | | |
| | FaceNet (VGGFace2) | | | | FaceNet (Webface) | | | | FaceNet (Webface) | | | | ArcFace | | | |
| (↓) | Mean | AAD | MAD | STD | Mean | AAD | MAD | STD | Mean | AAD | MAD | STD | Mean | AAD | MAD | STD |
|---|---|---|---|---|---|---|---|---|---|---|---|---|---|---|---|---|
| Baseline | 6.37 | 2.89 | 5.73 | 3.77 | 5.55 | 2.48 | 4.97 | 2.91 | 6.77 | 3.63 | 5.96 | 4.03 | 2.57 | 1.39 | 2.94 | 1.63 |
| AGENDA | 7.71 | 3.11 | 6.09 | 3.86 | 5.71 | 2.37 | 4.28 | 2.85 | 13.21 | 6.37 | 12.91 | 7.55 | 5.14 | 2.48 | 5.92 | 3.04 |
| PASS | 8.09 | 2.40 | 4.10 | 2.83 | 7.65 | 3.36 | 5.34 | 3.85 | 13.16 | 5.25 | 9.58 | 6.12 | 3.69 | 2.01 | 4.24 | 2.37 |
| FTC | 5.69 | 2.32 | 4.51 | 2.95 | 4.73 | 1.93 | 3.86 | 2.28 | 6.64 | 2.80 | 5.61 | 3.27 | 2.95 | 1.48 | 3.03 | 1.74 |
| GST | 2.34 | 0.82 | 1.58 | 0.98 | 3.24 | 1.21 | 1.93 | 1.34 | 3.09 | 1.45 | 2.80 | 1.65 | 3.34 | 1.81 | 4.21 | 2.19 |
| FSN | 1.43 | 0.35 | 0.57 | 0.40 | 2.49 | 0.84 | 1.19 | 0.91 | **2.76** | 1.38 | 2.67 | 1.60 | 2.65 | 1.45 | 3.23 | 1.71 |
| **FairCal (Ours)** | **1.37** | **0.28** | **0.50** | **0.34** | **1.75** | **0.41** | **0.64** | **0.45** | 3.09 | **1.34** | **2.48** | **1.55** | **2.49** | **1.30** | **2.68** | **1.52** |
| *Oracle (Ours)* | *1.18* | *0.28* | *0.53* | *0.33* | *1.35* | *0.38* | *0.66* | *0.43* | *2.23* | *1.15* | *2.63* | *1.40* | *1.41* | *0.59* | *1.30* | *0.69* |

Table 4: **Predictive equality**: For two choices of global FPR (two blocks of rows): 0.1% and 1%, we compare the deviations in subgroup FPRs in terms of: Average Absolute Deviation (AAD), Maximum Absolute Deviation (MAD), and Standard Deviation (STD). (lower is better in all cases)

| | | RFW | | | | | | BFW | | | | | |
| | | FaceNet (VGGFace2) | | | FaceNet (Webface) | | | FaceNet (Webface) | | | ArcFace | | |
| | (↓) | AAD | MAD | STD | AAD | MAD | STD | AAD | MAD | STD | AAD | MAD | STD |
|---|---|---|---|---|---|---|---|---|---|---|---|---|---|
| 0.1% FPR | Baseline | 0.10 | 0.15 | 0.10 | 0.14 | 0.26 | 0.16 | 0.29 | 1.00 | 0.40 | 0.12 | 0.30 | 0.15 |
| | AGENDA | 0.11 | 0.20 | 0.13 | 0.12 | 0.23 | 0.14 | 0.14 | 0.40 | 0.18 | **0.09** | 0.23 | **0.11** |
| | PASS | 0.11 | 0.18 | 0.12 | 0.11 | 0.18 | 0.12 | 0.36 | 1.21 | 0.49 | 0.12 | 0.29 | 0.14 |
| | FTC | 0.10 | 0.15 | 0.11 | 0.12 | 0.23 | 0.14 | 0.24 | 0.74 | 0.32 | **0.09** | **0.20** | **0.11** |
| | GST | 0.13 | 0.24 | 0.15 | **0.09** | **0.16** | **0.10** | 0.13 | 0.35 | 0.16 | 0.10 | 0.24 | 0.12 |
| | FSN | 0.10 | 0.18 | 0.11 | 0.11 | 0.23 | 0.13 | **0.09** | 0.20 | **0.11** | 0.11 | 0.28 | 0.14 |
| | **FairCal (Ours)** | **0.09** | **0.14** | **0.10** | **0.09** | **0.16** | **0.10** | **0.09** | **0.20** | 0.11 | 0.11 | 0.31 | 0.15 |
| | *Oracle (Ours)* | *0.11* | *0.19* | *0.12* | *0.11* | *0.20* | *0.13* | *0.12* | *0.25* | *0.15* | *0.12* | *0.27* | *0.14* |
| 1% FPR | Baseline | 0.68 | 1.02 | 0.74 | 0.67 | 1.23 | 0.79 | 2.42 | 7.48 | 3.22 | 0.72 | 1.51 | 0.85 |
| | AGENDA | 0.73 | 1.14 | 0.81 | 0.73 | 1.08 | 0.78 | 1.21 | 3.09 | 1.51 | 0.65 | 1.78 | 0.84 |
| | PASS | 0.89 | 1.52 | 1.01 | 0.68 | 0.99 | 0.73 | 3.30 | 10.18 | 4.34 | 0.72 | 2.00 | 0.93 |
| | FTC | 0.60 | 0.91 | 0.66 | 0.54 | 1.05 | 0.66 | 1.94 | 5.74 | 2.57 | 0.54 | **1.04** | 0.61 |
| | GST | 0.52 | 0.92 | 0.60 | 0.30 | 0.57 | 0.37 | 1.05 | 3.01 | 1.38 | **0.44** | 1.13 | **0.56** |
| | FSN | 0.37 | 0.68 | 0.46 | 0.35 | 0.61 | 0.40 | 0.87 | 2.19 | 1.05 | 0.55 | 1.27 | 0.68 |
| | **FairCal (Ours)** | **0.28** | **0.46** | **0.32** | **0.29** | **0.57** | **0.35** | **0.80** | **1.79** | **0.95** | 0.63 | 1.46 | 0.78 |
| | *Oracle (Ours)* | *0.40* | *0.69* | *0.45* | *0.41* | *0.74* | *0.48* | *0.77* | *1.71* | *0.91* | *0.83* | *2.08* | *1.07* |

It is important to emphasize that **accuracy** and **predictive equality** metrics are determined at a specified global FPR. This entails determining for each prior method a threshold that achieves the desired global FPR. However, this itself promotes our method's advantages: (a) previous methods such as FSN rely on a predefined FPR, while ours does not; (b) while other methods need recomputation for different thresholds, our method simultaneously removes the bias at multiple global FPRs.

## 6.1 GLOBAL ACCURACY

The receiver operating characteristic (ROC) curve plots the true positive rate (TPR = 1−FNR) against the FPR, obtained by thresholding the model at different values. The area under the ROC curve (AUROC) is thus a holistic metric that summarizes the **accuracy** of the classifiers (Davis & Goadrich, 2006). A higher AUROC is better, an uninformative classifier has an AUROC of 50%.

Our FairCal method achieves SOTA results, as shown in Table 2. We report the AUROC for the different pre-trained models and datasets, as well as the TPRs at 0.1% and 1% global FPR thresholds. FairCal achieves the best values of AUROC in all cases, the highest TPR in seven of the eight cases, and surpasses GST and our Oracle method (which use **subgroup information**) on the RFW dataset.

This overall **accuracy** improvement of our FairCal method can be explained as follows: the feature vectors contain more information than the score alone, so they can be used to identify pairs where the probability of an error is higher or lower, allowing per-cluster calibration to give better **accuracy**.

In our FairCal method, the subgroups are formed based on the features of the model, and not on manual human-assigned attributes. This allows for potentially more complex and effective subgroups. We visualized the cluster images, and found they indeed have semantic meaning, see Appendix A.

## 6.2 FAIRNESS CALIBRATION

The prior relevant methods (Baseline, AGENDA, PASS, FTC, GST, and FSN) do not output probabilities, hence by design they cannot be **fairly-calibrated**. However, in order to fully demonstrate that our method is superior to those approaches, we apply beta calibration (the same post-hoc calibration method used in our FairCal method) to their score outputs.

Calibration error is a measure of the deviation in a model's output probabilities of matches from the actual results. When considering subgroups, this is measured by two quantities: (i) the calibration error on each subgroup (lower is better, since it means fewer errors), which can be measured using the Brier score (BS) (DeGroot & Fienberg, 1983), Expected Calibration Error (ECE) (Guo et al., 2017), or Kolmogorov-Smirnov (KS) (Gupta et al., 2021) (discussed in Appendix E). We present results for KS, which has been established to be the best measure (Gupta et al., 2021). (ii) The deviation in these quantities across subgroups (lower is better, since it is more fair), measured as: Average Absolute Deviation (AAD), Maximum Absolute Deviation (MAD), and Standard Deviation (STD).

Our FairCal method achieves the best results, as shown in Table 3: the best mean in three of the four cases (smaller errors) and the best deviation in nine of the nine cases (more fair across subgroups). The improvement is most significant on the less **accurate** models. We discuss these results in more detail in the Appendix.

## 6.3 PREDICTIVE EQUALITY

As **predictive equality** is achieved through equal FPRs between different subgroups, we can measure bias by quantifying the deviation of these FPRs. We report three measures of deviation (AAD, MAD, and STD) at two different global FPRs: 0.1% and at 1.0%.

Our FairCal method achieves the best results, as shown in Table 4: the best or tied measure of **predictive equality** in 18 of the 24 cases. In the cases where FairCal is not best, i.e., when the ArcFace model evaluated on the BFW dataset, FTC and GST provide the best results, but the differences between AGENDA's, PASS's, FTC's, GST's, FSN's, and FairCal's deviation measures are within a fraction of 1%. Moreover, when applied to ArcFace, the FTC and GST methods reduce the model's **accuracy**.

## 7 CONCLUSION

We introduce FairCal, a post-training calibration method that (i) achieves **SOTA accuracy**, (ii) is **fairly-calibrated**, (iii) achieves **predictive equality (equal FPRs)** on challenging face verification datasets, (iv) **without the knowledge of any sensitive attribute**. It can be readily applied to existing FR systems, as it is a statistical method that (v) **does not require any retraining**. It can aid human-users in using FR systems to make better informed decisions based on interpretable, unbiased results.

**Limitations**: Only under very strict conditions (e.g. a perfect classifier) can all three fairness notions (**fairness calibration**, **predictive equality**, equal opportunity) be satisfied. While we have shown that our FairCal method can achieve two out of the three definitions—which is the best we can hope for in practice—it requires the use of a calibration dataset. Deploying our method on a substantially different dataset would most likely require further calibration. Calibrating models for never-seen data is currently an open problem.

## ETHICS STATEMENT

Bolstered by dramatically improving accuracy, Facial recognition systems have exploded in popularity in recent years, and have been applied to innumerable settings under various use cases. However, the severe biases within these systems limit their applicability, and open the doors to widespread social harm. Our paper is primarily focused on addressing the lack of fairness as well as potential unethical uses of current face recognition systems, specifically in the context of face verification. We address this issue by proposing a novel post-processing method that significantly reduces the false positive rates across demographic subgroups without requiring knowledge of any sensitive attributes. At many places in the paper, we have referenced several prior works that mention and analyze the current and potential risks posed by current facial recognition systems. We identified the shortcomings in the previous methods, and methodically established a scientific way of identifying and measuring fairness in the current systems. Our experiments involve large public datasets of faces. Our work explicitly addresses problems with facial recognition systems that could misuse sensitive attributes in these datasets, such as ethnicity, race, gender, etc. The motivation of our method is in trying to mitigate bias and improve fairness in the systems using these datasets.

## REPRODUCIBILITY STATEMENT

In order to ensure the reproducibility of our work, we described our proposed methods, FairCal and Oracle, in detail in section 4. The details of our experiments, including the datasets and pre-processing steps, can be found in section 5. Additional details on how we implemented AGENDA and FTC, which require training additional models, are provided in Appendix B and Appendix D, respectively. The embeddings from the pretrained models were obtained on a machine with one GeForce GTX 1080 Ti GPU. All methods were implemented in Python, and the code is provided in the supplemental material.

## ACKNOWLEDGMENTS

This material is based upon work supported by the Air Force Office of Scientific Research under award number FA9550-18-1-0167 (A.O.) Thanks to CIFAR for their support through the CIFAR AI Chairs program. We would also like to thank Muawiz Chaudhary and Vicent Mai for their valuable feedback.

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

# Supplemental Material

## A    EXAMPLES OF THE UNSUPERVISED CLUSTERS

In order to not rely on the sensitive attribute like the Oracle method, our FairCal method uses unsupervised clusters computed with the $K$-means algorithm based on the feature embeddings of the images. We found them to have semantic meaning. Some examples are included in Figure 3.

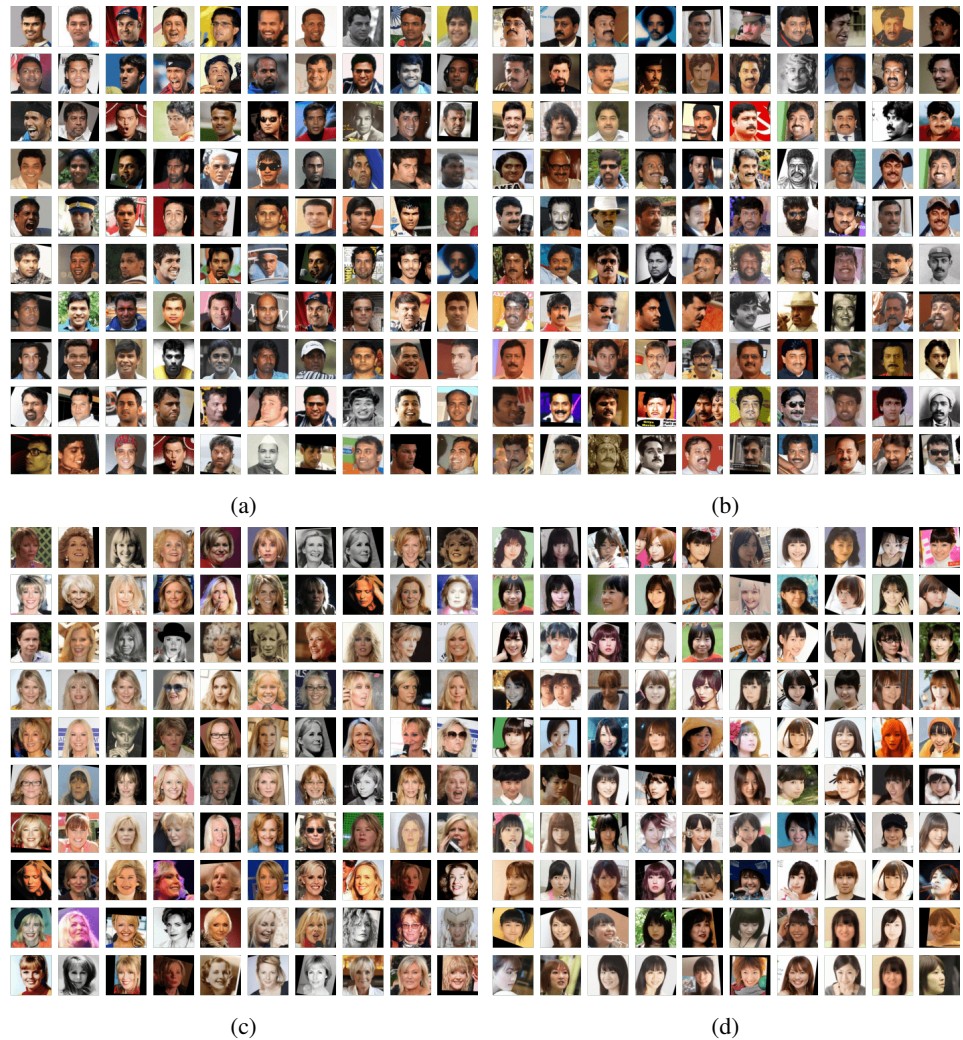

(a)                                                                                  (b)

(c)                                                                                  (d)

Figure 3: Examples of clusters obtained with the $K$-means algorithm ($k = 100$) on the RFW dataset based on the feature embeddings computed with the FaceNet model: (a) Indian men with no facial hair, (b) Indian men with moustaches, (c) Caucasian women with blond hair, (d) young Asian women with dark hair.

## B    AGENDA METHOD

The Adversarial Gender De-biasing algorithm (AGENDA) learns a shallow network that removes the gender information of the embeddings from a pre-trained network producing new embeddings. The algorithm entails training a generator model $M$, a classifier $C$ and a discriminator $E$. As proposed, the algorithm only removes the gender information using an adversarial loss that encourages the discriminator to produce equal probabilities for both male and female gender. Since the RFW dataset

contains the ethnicity attribute (as opposed to gender) and the BFW dataset contains both gender and ethnicity, we modify the loss to encourage the discriminator to produce equal probabilities amongst all possible sensitive attributes.

The shallow networks used for $M$, $C$ and $E$ are the same as the ones specified in Dhar et al. (2020), with the exception that $E$ has as many outputs as possible values of sensitive attributes (4 for RFW and 8 for BFW).

AGENDA requires the use of a validation set to determine if the discriminator should continue to be updated or not. Hence the embeddings used for training into a 80/20 split, with the 20 split used for the validation.

In Stage 1, the generator $M$ and classifier $C$ are trained for 50 epochs. Then the bulk of training consists of 100 episodes: (i) Stage 2 is repeated every 10 episodes and consists in training the discriminator $E$ for 25 epochs; (ii) in Stage 3 both $M$ and $C$ are trained with the adversarial loss for 5 epochs with $\lambda = 10$; (iii) in Stage 4 the discriminator is updated for 5 epochs, unless its accuracy on the validations set is higher than 90%. All training is done with a batch size of 400 and an ADAM optimizer with a learning rate of $10^{-3}$. For more details, we refer to the code provided in the supplemental material.

## C    PASS METHOD

Similar to AGENDA, PASS (Dhar et al., 2021) learns a shallow network to removes the sensitive information of the embeddings from a pre-trained network producing new embeddings. The algorithm entails training a generator model $M$, a classifier $C$ and ensemble of discriminator $E$.

The shallow networks used for $M$, $C$ and $E$ are the same as the ones specified in Dhar et al. (2021). Like AGENDA, it requires the use of a validation set. The same partition as in AGENDA was used.

In Stage 1, the generator $M$ and classifier $C$ are trained for 100 epochs. Then the bulk of training consists of 100 episodes: (i) Stage 2 is repeated every 10 episodes and consists in training the discriminator $E$ for 100 epochs; (ii) in Stage 3 both $M$ and $C$ are trained with the adversarial loss for 5 epochs with $\lambda = 10$; (iii) in Stage 4 we use the One-At-a-time strategy to update one of the discriminators in the ensemble $E$. It is updated for 5 epochs, unless its accuracy on the validations set is higher than 95%. All training is done with a batch size of 400 and an ADAM optimizer with a learning rate of $10^{-3}$.

The availability of sensitive attributes determined the choice of number of discriminators used. For the RFW dataset, since only the race is know, we use an ensemble with $K = 2$ discriminators. For the BFW, since both race and gender are known, we use an ensemble of $K = 2$ discriminators for the race sensitive attribute and $K = 3$ discriminators for the gender sensitive attribute. The latter in fact constitutes the MultiPASS variant of PASS. We choose to name it here PASS for simplicity. For more details, we refer to the code provided in the supplemental material.

## D    FAIR TEMPLATE COMPARISON (FTC) METHOD

The Fair Template Comparison (FTC) method Terhörst et al. (2020a) learns a shallow network with the goal of outputting fairer decisions. We implemented the FTC method as follow. In order to keep the ratios between the dimensions of layers the same as in the original paper Terhörst et al. (2020a), we used a 512-dimensional input layer, followed by two 2048-dimensional intermediate layers. The final layer is a fully connected linear layer with 2-dimensional output with a softmax activation. All intermediate layers are followed by a ReLU activation function and dropout (with $p = 0.3$). The network was trained with a batchsize of $b = 200$ over 50 epochs, using an Adam optimizer with a learning rate of $10^{-3}$ and weight decay of $10^{-3}$. Two losses, one based on subgroup fairness and the other on both subgroup and individual fairness, were proposed in Terhörst et al. (2020a). Based on the paper's recommendations, we used the individual fairness loss with a trade-off parameter of $\lambda = 0.5$.

## E    MEASURING CALIBRATION ERROR

There are different metrics available to measure if a probabilistic classifier is calibrated or fairly-calibrated. Calibration error is the error between the true and estimated confidences and is typically measured by the Expected Calibration Error (ECE) (Guo et al., 2017):

Despite being the most popular calibration error metric, the ECE has several weaknesses, chief among which is its dependence on the binning scheme Nixon et al. (2019). Recently, Gupta et al. (2021) introduced a simple, bin-free calibration measure. For calibrated scores $P(Y = 1|C = c) = c$ we have, by Bayes' rule:

$$P(Y = 1, C = c) = cP(C = c).$$

Inspired by the Kolmogorov-Smirnov (KS↓) statistic test, Gupta et al. (2021) proposed to measure the calibration error by comparing the cumulative distributions of $P(Y = 1, C = c)$ and $cP(C = c)$, which empirically correspond to computing the sequences

$$h_i = h_{i-1} + \mathbf{1}_{y_i=1}/N \quad \text{and} \quad \tilde{h}_i = \tilde{h}_{i-1} + c_i/N$$

with $h_0 = \tilde{h}_0 = 0$, and $N$ is the total number of samples. Then the KS calibration error metric is given by

$$KS = \max_i \left| h_i - \tilde{h}_i \right|.$$

Another measure is the Brier score (BS↓) (DeGroot & Fienberg, 1983), which estimates the mean squared error between the correctness of prediction and the confidence score:

$$BS(\mathcal{D}) = \frac{1}{|\mathcal{D}|} \sum_{(c_i,y_i)\in\mathcal{D}} \left(\mathbf{1}_{\hat{y}_i=y_i} - c_i\right)^2 \tag{5}$$

For all the above metrics (ECE, KS, BS), lower is better.

## F    FAIRNESS CALIBRATION

Since the calibration map produced by beta calibration is monotone, the ordering of the images provided by the scores is the same as the ordering provided by the probabilities; therefore, the **accuracy** of the methods wheh thresholding remains unchanged. The calibration error (CE) measured with an adaptation of the Kolmogorov-Smirnov (KS) test (described in Appendix E) is computed for each subgroup of interest. Notice that for the BFW dataset we consider the eight subgroups that result from the intersection of the ethnicity and gender subgroups.

We first observe that all methods are equally globally calibrated (i.e., the calibration error is low) after the post-hoc calibration method is applied, except for the FTC on the RFW dataset (see the Global column in Table 5 and Table 6).

By inspecting Table 5 and Table 6, we notice that, after calibration, the Baseline method results in models that are not fairly-calibrated, though perhaps not in the way one would expect. Typically, bias is directed against minority groups, but in this case, it is the Caucasian subgroups that have the higher CEs. This is a consequence of the models' above average accuracy on this subgroup, which is underestimated and therefore not captured by the calibration procedure. It is important to point out that this is not a failure of the calibration procedure, since the global CE (i.e., the CE measured on all pairs) is low, as discussed above.

## G    EQUAL OPPORTUNITY (EQUAL FNR)

While equal opportunity (equal FNRs between subgroups) is not prioritized for the FR systems when used by law enforcement, it may be prioritized in different contexts such as office building security. Empirically, our method also mitigates the equal opportunity bias at low global FNRs, as can be seen in Table 7.

Table 5: KS on all the pairs (Global (Gl)) and on each ethnicity subgroup (African (Af), Asian (As), Caucasian (Ca), Indian (In) using beta calibration on the RFW dataset.

| (↓) | FaceNet (VGGFace2) | | | | | FaceNet (Webface) | | | | |
|---|---|---|---|---|---|---|---|---|---|---|
| | Gl | Af | As | Ca | In | Gl | Af | As | Ca | In |
| Baseline | 0.78 | 6.16 | 5.74 | 12.06 | 1.53 | 0.69 | 3.89 | 4.34 | 10.52 | 3.46 |
| AGENDA | 1.02 | 3.66 | 6.97 | 13.76 | 6.46 | 1.21 | 1.75 | 4.94 | 9.39 | 6.77 |
| PASS | 0.81 | 7.08 | 8.97 | 12.01 | 4.32 | 0.81 | 8.06 | 6.11 | 12.99 | 3.43 |
| FTC | 1.12 | 5.13 | 5.41 | 10.19 | 2.02 | 1.25 | 3.19 | 3.81 | 8.59 | 3.35 |
| GST | 0.77 | 1.99 | 3.66 | 2.30 | 1.40 | 0.71 | 2.01 | 4.97 | 3.83 | 2.16 |
| FSN | 0.77 | 1.27 | 1.62 | 1.49 | 1.35 | 0.85 | 1.94 | 3.06 | 1.70 | 3.27 |
| **FairCal (Ours)** | 0.81 | 1.11 | 1.41 | 1.29 | 1.70 | 0.70 | 1.48 | 1.62 | 1.68 | 2.21 |
| *Oracle (Ours)* | *0.76* | *0.99* | *1.28* | *1.2* | *1.25* | *0.62* | *1.54* | *1.46* | *1.13* | *1.25* |

Table 6: KS on all the pairs (Global (Gl)) and on each ethnicity and gender subgroup (African Females (AfF), African Males (AfM), Asian Females (AsF), Asian Males (AsM), Caucasian Females (CF), Caucasian Males (CM), Indian Females (IF), Indian Males (IM)) using beta calibration on the BFW dataset.

| (↓) | FaceNet (Webface) | | | | | | | | | ArcFace | | | | | | | | |
|---|---|---|---|---|---|---|---|---|---|---|---|---|---|---|---|---|---|---|
| | Gl | AfF | AfM | AsF | AsM | CF | CM | IF | IM | Gl | AfF | AfM | AsF | AsM | CF | CM | IF | IM |
| Baseline | 0.48 | 5.00 | 2.17 | 11.19 | 2.93 | 12.06 | 10.41 | 5.58 | 4.80 | 0.37 | 1.52 | 3.17 | 5.30 | 4.28 | 1.31 | 1.10 | 2.09 | 1.81 |
| AGENDA | 1.44 | 13.49 | 12.66 | 5.64 | 8.52 | 25.18 | 23.43 | 5.72 | 11.06 | 0.99 | 5.39 | 5.44 | 11.07 | 7.91 | 2.66 | 2.26 | 3.33 | 3.09 |
| PASS | 0.95 | 11.02 | 13.80 | 12.93 | 11.58 | 21.21 | 16.50 | 12.74 | 5.49 | 0.72 | 3.66 | 4.47 | 7.72 | 6.25 | 1.56 | 0.94 | 2.76 | 2.16 |
| FTC | 0.56 | 7.33 | 4.06 | 5.71 | 3.68 | 12.25 | 10.47 | 4.13 | 5.51 | 0.49 | 2.02 | 3.56 | 5.77 | 4.62 | 1.80 | 1.03 | 2.65 | 2.15 |
| GST | 0.40 | 1.62 | 3.23 | 4.52 | 4.61 | 1.32 | 2.34 | 4.36 | 2.72 | 0.48 | 2.38 | 3.50 | 7.46 | 5.03 | 0.99 | 1.59 | 3.36 | 2.44 |
| FSN | 0.39 | 2.35 | 3.12 | 4.16 | 4.40 | 1.50 | 0.99 | 3.54 | 2.02 | 0.38 | 1.74 | 3.01 | 5.70 | 4.30 | 1.02 | 1.15 | 2.45 | 1.81 |
| **FairCal (Ours)** | 0.59 | 3.83 | 2.55 | 2.92 | 3.79 | 3.70 | 2.43 | 3.21 | 2.32 | 0.49 | 1.73 | 3.12 | 4.79 | 3.81 | 1.05 | 1.16 | 2.28 | 1.97 |
| *Oracle (Ours)* | *0.43* | *1.67* | *2.3* | *2.83* | *2.49* | *0.67* | *1.24* | *4.6* | *2.02* | *0.32* | *1.26* | *0.99* | *1.93* | *1.72* | *0.86* | *1.15* | *1.64* | *1.74* |

Table 7: **Equal opportunity**: Each block of rows represents a choice of global FNR: 0.1% and 1%. For a fixed a global FNR, compare the deviations in subgroup FNRs in terms of three deviation measures: Average Absolute Deviation (AAD), Maximum Absolute Deviation (MAD), and Standard Deviation (STD) (lower is better).

| | (↓) | RFW | | | | | | BFW | | | | | |
|---|---|---|---|---|---|---|---|---|---|---|---|---|---|
| | | FaceNet (VGGFace2) | | | FaceNet (Webface) | | | FaceNet (Webface) | | | ArcFace | | |
| | | AAD | MAD | STD | AAD | MAD | STD | AAD | MAD | STD | AAD | MAD | STD |
| | Baseline | 0.09 | 0.13 | 0.10 | 0.10 | 0.16 | 0.11 | 0.09 | 0.23 | 0.11 | 0.11 | 0.31 | 0.14 |
| | AGENDA | 0.11 | 0.22 | 0.13 | 0.10 | 0.14 | 0.11 | 0.14 | 0.34 | 0.16 | 0.09 | 0.24 | 0.12 |
| | PASS | 0.09 | 0.14 | 0.10 | 0.11 | 0.20 | 0.12 | 0.10 | 0.33 | 0.14 | 0.10 | 0.31 | 0.14 |
| 0.1% FNR | FTC | 0.09 | **0.11** | **0.09** | **0.08** | 0.14 | **0.1** | **0.04** | **0.09** | **0.05** | **0.06** | **0.14** | **0.07** |
| | GST | 0.09 | 0.13 | 0.10 | 0.12 | 0.21 | 0.13 | 0.10 | 0.26 | 0.12 | 0.12 | 0.37 | 0.15 |
| | FSN | **0.09** | 0.13 | 0.09 | 0.09 | **0.14** | 0.10 | 0.07 | 0.22 | 0.10 | 0.12 | 0.33 | 0.15 |
| | **FairCal (Ours)** | 0.10 | 0.14 | 0.10 | 0.11 | 0.17 | 0.12 | 0.10 | 0.27 | 0.13 | 0.09 | 0.17 | 0.10 |
| | *Oracle (Ours)* | *0.11* | *0.18* | *0.12* | *0.12* | *0.21* | *0.13* | *0.09* | *0.24* | *0.11* | *0.11* | *0.32* | *0.14* |
| | Baseline | 0.60 | 0.96 | 0.67 | 0.45 | 0.81 | 0.53 | 0.39 | 0.84 | 0.47 | 0.75 | 1.85 | 0.93 |
| | AGENDA | 0.99 | 1.97 | 1.16 | 0.67 | 1.33 | 0.81 | 0.90 | 2.39 | 1.15 | 0.72 | 1.54 | 0.84 |
| | PASS | 0.77 | 1.06 | 0.81 | 0.83 | 1.64 | 0.99 | 0.72 | 1.79 | 0.89 | 0.71 | 1.76 | 0.90 |
| 1% FNR | FTC | 0.48 | 0.83 | 0.56 | **0.32** | **0.58** | **0.38** | **0.3** | **0.62** | **0.34** | **0.49** | **1.12** | **0.6** |
| | GST | 0.39 | 0.60 | 0.44 | 0.54 | 0.96 | 0.62 | 0.49 | 1.02 | 0.57 | 0.83 | 2.27 | 1.07 |
| | FSN | **0.28** | **0.47** | **0.32** | 0.40 | 0.78 | 0.48 | 0.41 | 0.92 | 0.49 | 0.77 | 1.91 | 0.96 |
| | **FairCal (Ours)** | 0.30 | 0.51 | 0.34 | 0.39 | 0.72 | 0.48 | 0.32 | 0.74 | 0.40 | 0.65 | 1.48 | 0.80 |
| | *Oracle (Ours)* | *0.38* | *0.61* | *0.42* | *0.56* | *1.06* | *0.67* | *0.37* | *0.77* | *0.44* | *0.5* | *1.11* | *0.6* |

Table 8: **Fairness calibration:** measured by the mean KS across the sensitive subgroups on the IJB-C dataset. **Bias:** measured by the deviations of KS across subgroups in terms of three deviation measures: Average Absolute Deviation (AAD), Maximum Absolute Deviation (MAD), and Standard Deviation (STD). (Lower is better in all cases.)

| (↓) | FaceNet (VGGFace2) | | | | FaceNet (Webface) | | | |
|---|---|---|---|---|---|---|---|---|
| | Mean | AAD | MAD | STD | Mean | AAD | MAD | STD |
| Baseline | 3.54 | 1.90 | 5.46 | 2.38 | 3.57 | 2.01 | 4.60 | 2.37 |
| AGENDA | 2.61 | 1.31 | 3.82 | 1.65 | 3.41 | 1.28 | 4.33 | 1.76 |
| PASS | 7.22 | 3.24 | 8.40 | 3.98 | 6.57 | 3.50 | 9.12 | 4.25 |
| FTC | 2.92 | 1.44 | 3.98 | 1.80 | 2.98 | 1.41 | 3.43 | 1.69 |
| GST | 2.38 | 1.36 | 5.09 | 1.88 | 3.52 | 1.59 | 6.88 | 2.42 |
| FSN | 2.12 | 1.18 | 3.74 | 1.53 | 2.45 | 1.05 | 3.49 | 1.42 |
| **FairCal (Ours)** | **1.8** | **0.95** | **2.81** | **1.2** | **2.12** | **0.99** | **2.51** | **1.18** |
| *Oracle (Ours)* | *1.84* | *1.05* | *4.32* | *1.53* | *1.87* | *0.91* | *3.23* | *1.27* |

Table 9: Global **accuracy** (higher is better) measured by AUROC, and TPR at two FPR thresholds on the IJB-C dataset.

| (↑) | FaceNet (VGGFace2) | | | FaceNet (Webface) | | |
|---|---|---|---|---|---|---|
| | AUROC | TPR @ 0.1% FPR | TPR @ 1% FPR | AUROC | TPR @ 0.1% FPR | TPR @ 1% FPR |
| Baseline | 92.72 | 43.53 | 62.80 | 89.23 | 19.50 | 50.10 |
| AGENDA | 89.86 | 44.78 | 61.65 | 81.33 | 32.90 | 46.71 |
| PASS | 86.68 | 24.06 | 39.46 | 81.54 | 9.89 | 28.66 |
| FTC | 91.60 | 39.41 | 60.96 | 87.64 | 17.70 | 47.24 |
| GST | 92.78 | 47.65 | 66.34 | 89.13 | 25.06 | 50.57 |
| FSN | 92.48 | **53.4** | 68.29 | 88.00 | **44.74** | 58.52 |
| **FairCal (Ours)** | **94.74** | 52.54 | **68.45** | **92.04** | 44.28 | **58.98** |
| *Oracle (Ours)* | *93.18* | *48.4* | *66.96* | *89.8* | *24.13* | *53.62* |

## H  IJB-C RESULTS

We report additional results in Table 9, Table 8 and Table 10. on the IJB-C dataset (Maze et al., 2018) following the same guidelines as in the RFW and BFW datasets. We include results for the FaceNet (VGGFace2) and FaceNet (Webface) models and they are in agreement with the results presented in the main paper on RFW and BFW. (We do not include results for the ArcFace model as we were unable to overcome software issues to compute its embeddings).

## I  STANDARD POST-HOC CALIBRATION METHODS

For completeness, we provide a brief description of the post-hoc calibration methods used in this work. Beta calibration Kull et al. (2017) was used to obtain our main results, but we show below that choosing others methods (Histogram Binning (Zadrozny & Elkan, 2001) or Isotonic Regression (Zadrozny & Elkan, 2001; Niculescu-Mizil & Caruana, 2005)) does not impact the performance of our FairCal method.

Throughout this section, we denote by $\mathcal{P}^{\mathrm{cal}}$ the pairs of images in the calibration set and $S^{\mathrm{cal}}$ their respective cosine similarity scores such that

$$S^{\mathrm{cal}} = \{s(\hat{\boldsymbol{x}}_1, \hat{\boldsymbol{x}}_2) : (\hat{\boldsymbol{x}}_1, \hat{\boldsymbol{x}}_2) \in \mathcal{P}^{\mathrm{cal}}\}.$$

Table 10: **Predictive equality**: For two choices of global FPR (two blocks of rows): 0.1% and 1%, we compare, on the IJB-C dataset, the deviations in subgroup FPRs in terms of: Average Absolute Deviation (AAD), Maximum Absolute Deviation (MAD), and Standard Deviation (STD). (lower is better in all cases)

|  | | FaceNet (VGGFace2) | | | FaceNet (Webface) | | |
|---|---|---|---|---|---|---|---|
|  | | AAD | MAD | STD | AAD | MAD | STD |
| 0.1% FPR | Baseline | 0.10 | 0.33 | 0.13 | 0.11 | 0.51 | 0.17 |
|  | AGENDA | 0.09 | 0.28 | 0.12 | 0.09 | 0.37 | 0.13 |
|  | PASS | 0.12 | 0.48 | 0.17 | 0.13 | 0.65 | 0.21 |
|  | FTC | 0.10 | 0.33 | 0.13 | 0.10 | 0.47 | 0.16 |
|  | GST | 0.08 | 0.33 | 0.12 | 0.07 | 0.24 | 0.10 |
|  | FSN | **0.06** | **0.24** | **0.09** | 0.06 | 0.20 | 0.08 |
|  | **FairCal (Ours)** | **0.06** | 0.25 | **0.09** | **0.05** | **0.13** | **0.06** |
|  | *Oracle (Ours)* | *0.06* | *0.26* | *0.1* | *0.08* | *0.27* | *0.11* |
| 1% FPR | Baseline | 0.93 | 3.15 | 1.23 | 0.85 | 3.41 | 1.21 |
|  | AGENDA | 0.65 | 2.04 | 0.85 | 0.63 | 2.60 | 0.94 |
|  | PASS | 1.10 | 3.69 | 1.45 | 1.16 | 4.94 | 1.72 |
|  | FTC | 0.90 | 2.88 | 1.15 | 0.81 | 3.13 | 1.14 |
|  | GST | 0.60 | 2.64 | 0.92 | 0.68 | 2.51 | 0.95 |
|  | FSN | 0.52 | 1.85 | 0.70 | 0.49 | 1.65 | 0.65 |
|  | **FairCal (Ours)** | **0.47** | **1.72** | **0.66** | **0.44** | **1.33** | **0.56** |
|  | *Oracle (Ours)* | *0.46* | *1.85* | *0.69* | *0.47* | *1.52* | *0.64* |

### I.1 HISTOGRAM BINNING

In Histogram Binning (Zadrozny & Elkan, 2001), we partition $S^{\text{cal}}$ into $m$ bins $B_i$ with equal-mass, where $i = 1, \ldots, m$. Then, given a pair of images $(\boldsymbol{x}_1, \boldsymbol{x}_2)$ with score $s(\boldsymbol{x}_1, \boldsymbol{x}_2) \in B_i$, we define

$$c(\boldsymbol{x}_1, \boldsymbol{x}_2) = \frac{1}{|B_i|} \sum_{\substack{s(\hat{\boldsymbol{x}}_1, \hat{\boldsymbol{x}}_2) \in B_i \\ (\boldsymbol{x}_1, \boldsymbol{x}_2) \in \mathcal{P}^{\text{cal}}}} \mathbf{1}_{I(\hat{\boldsymbol{x}}_1) = I(\hat{\boldsymbol{x}}_2)}. \tag{6}$$

In other words, we simply count the number of scores in each bin that correspond to genuine pairs of images, i.e., images that belong to the same person. By construction, a confidence score $c$ (Equation 6) satisfies the binned version of the standard calibration (Definition 1). As for the bins, they can be chosen so as to have equal mass or to be equally spaced, or else by maximizing mutual information, as recently proposed in Patel et al. (2021). In this work, we created bins with equal mass.

Despite being an extremely computationally efficient method and providing good calibration, histogram binning is not guaranteed to preserve the monotonicity between scores and confidences, which is typically a desired property. Monotonicity ensures that the accuracy of the classifier is the same when thresholding either the scores or the calibrated confidences.

### I.2 ISOTONIC REGRESSION

Isotonic Regression (Zadrozny & Elkan, 2001; Niculescu-Mizil & Caruana, 2005) learns a monotonic function $\mu : \mathbb{R} \to \mathbb{R}$ by solving

$$\arg\min_{\mu} \frac{1}{|\mathcal{P}^{\text{cal}}|} \sum_{(\boldsymbol{x}_1, \boldsymbol{x}_2) \in \mathcal{P}^{\text{cal}}} \left( \mu(s(\boldsymbol{x}_1, \boldsymbol{x}_2)) - \mathbf{1}_{I(\hat{\boldsymbol{x}}_1) = I(\hat{\boldsymbol{x}}_2)} \right)^2$$

The confidence score is then given by $c(\boldsymbol{x}_1, \boldsymbol{x}_2) = \mu(s(\boldsymbol{x}_1, \boldsymbol{x}_2))$.

### I.3 Beta Calibration

Beta calibration (Patel et al., 2021) is a parametric calibration method, which learns a calibration map $\mu : \mathbb{R} \to \mathbb{R}$ of the form

$$c_{\boldsymbol{\theta}}(s) = \mu(s; \theta_1, \theta_2, \theta_3) = \frac{1}{1 + 1/\left(e^{\theta_3} \frac{s^{\theta_1}}{(1-s)^{\theta_2}}\right)}$$

where the parameters $\boldsymbol{\theta} = (\theta_1, \theta_2, \theta_3) \in \mathbb{R}^3$ are chosen by minimizing the log-loss function

$$LL(c, y) = y(-\log(c)) + (1-y)(-\log(1-c))$$

where $c = \mu(s(\boldsymbol{x}_1, \boldsymbol{x}_2))$. By restricting, $a$ and $b$ to be positive, the calibration map is monotone.

## J Robustness of FairCal results to hyperparameters

In this section we show that the results presented in the main paper still hold if we vary model hyperparameters, such as the number $K$ of clusters used in FairCal, and the calibration method.

**Choice of post-hoc calibration**: The implementation of the FairCal method requires choosing a post-hoc calibration method and the number of clusters $K$ in the $K$-means algorithm. Our method is robust to the choice of both with respect to fairness-calibration (the metric of interest when it comes to calibration) and its bias as depicted in Figure 4, Figure 5, Figure 6 and Figure 7. We compare with binning and isotonic regression. For the former, we chose 10 and 25 bins for the RFW and BFW datasets, respectively, given the different number of pairs in each dataset.

**Comparison to FSN (Terhörst et al., 2020b)**: The improved performance of FairCal over FSN is consistent across different choices of $K$. Fixing the choice of the post-hoc calibration method as beta calibration as in the results in the paper, we compare the two, together with Baseline and Oracle for additional baselines. Results are displayed in Figure 8, Figure 9, Figure 10, Figure 11, Figure 12, Figure 13 and Figure 14

## K Extending FairCal to multi-class classification

Consider a probabilistic classifier $\boldsymbol{p} : \mathcal{X} \to \Delta_k$ that outputs that outputs class probabilities for $k$ classes $1, \ldots, k$. Here $\mathcal{X}$ denotes the input space and $\Delta_k = \{(q_1, \ldots, q_k) : \sum_{i=1}^{k} q_k = 1\}$. Suppose $\boldsymbol{p}$ is computed with a deep neural network where $f$ denotes its feature encoder and $g$ the final fully connected layer. Then we write $\boldsymbol{p} = \text{softmax}(g(f(\boldsymbol{x})))$. Post-hoc calibration methods (like Dirichlet calibration (Kull et al., 2019), the generalization of beta calibration) compute a calibration map $\boldsymbol{\mu} : \Delta_k \to \Delta_k$ using a calibration set $\mathcal{X}^{\text{cal}} \subseteq \mathcal{X}$ such that $\boldsymbol{\mu} \circ \boldsymbol{p}$ is calibrated.

FairCal can be extended to multi-class classification as follows:

(i) Compute $\mathcal{Z}^{\text{cal}} = \{f(\boldsymbol{x}) : \boldsymbol{x} \in \mathcal{X}^{\text{cal}}\}$.

(ii) Apply the $K$-means algorithm to the image features, $\mathcal{Z}^{\text{cal}}$, partitioning the embedding space $\mathcal{Z}$ into $K$ clusters $\mathcal{Z}_1, \ldots, \mathcal{Z}_k$. These form the $K$ calibration sets:

$$\mathcal{X}_k^{\text{cal}} = \{\boldsymbol{x} : f(\boldsymbol{x}) \in \mathcal{Z}_k\}, \quad k = 1, \ldots, K \tag{7}$$

(iii) For each calibration set $\mathcal{X}_k^{\text{cal}}$, use a post-hoc calibration method to compute the calibration map $\boldsymbol{\mu}_k$.

(iv) For a new input $\boldsymbol{x}$ find the clusters it belongs to and output $\mu_k(\boldsymbol{p}(\boldsymbol{x}))$.

Notice that in the multi-class classification setting, we are no longer dealing with pairs and therefore each input (now a single image) belongs to one and only one cluster, thus simplifying the final setup. We leave for future work its implementation and study through a thorough empirical evaluation.

## L    RESULTS PRESENTED WITH STANDARD DEVIATIONS

Recall that the results presented in the main text were computed by taking the mean of a 5-fold leave-one-out cross-validation. Below, we report the corresponding standard deviations of the five folds. The standard deviations for the results on **accuracy** reported in Table 2 can be found in Table 11, Table 12, Table 13. For fairness-calibration in Table 3, they can be found in Table 14, Table 15,Table 16, Table 17. Finally, for predictive equality and equal opportunity in Table 4 and Table 7, they can be found in Table 18, Table 19, Table 20 and Table 21, Table 22, Table 23.

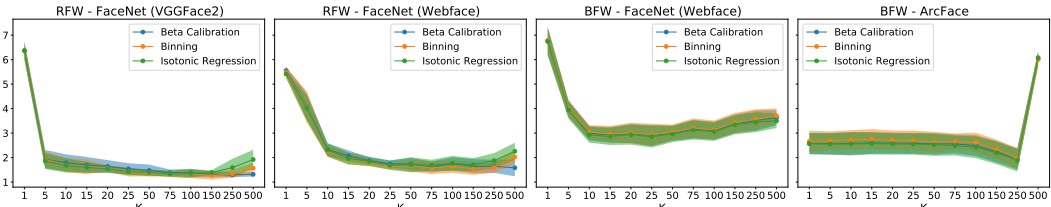

Figure 4: Comparison of **fairness-calibration** as measured by the subgroup mean of the KS across the sensitive subgroups for different values of $K$ and different choices of post-hoc calibration methods. Shaded regions refer to the standard error across the 5 different folds in the datasets.

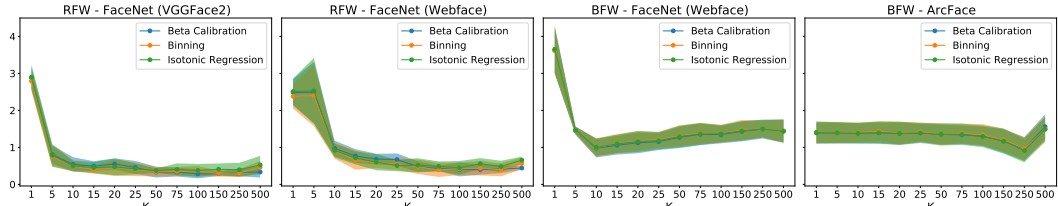

Figure 5: Bias in **fairness-calibration** as measured by the AAD (Average Absolute Deviation) in the KS across the sensitive subgroups for different values of $K$ and different choices of post-hoc calibration methods. Shaded regions refer to the standard error across the 5 different folds in the datasets.

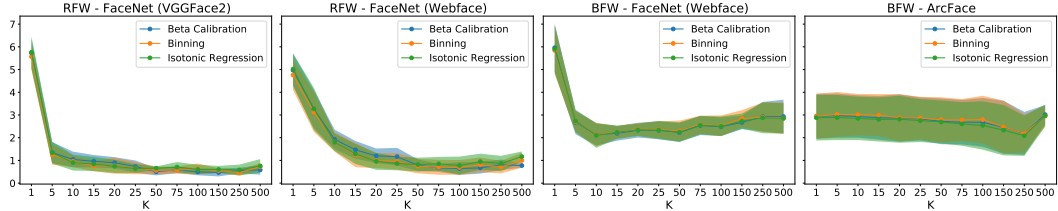

Figure 6: Bias in **fairness-calibration** as measured by the MAD (Maximum Absolute Deviation) in the KS across the sensitive subgroups for different values of $K$ and different choices of post-hoc calibration methods. Shaded regions refer to the standard error across the 5 different folds in the datasets.

## M    ADDITIONAL FIGURES

In Figure 1, we provided a qualitative analysis of the improved fairness/reduction in bias in terms of predictive equality on the RFW dataset with the FaceNet (Webface) model. We replicate this figure for the FaceNet (VGGFace2) model on the RFW dataset in Figure 15, and on the BFW dataset for the FaceNet (Webface) model in Figure 16 and ArcFace model in Figure 17.

Table 11: Global **accuracy** measured by the AUROC.

| (↑) | RFW | | BFW | |
| --- | --- | --- | --- | --- |
| | FaceNet (VGGFace2) | FaceNet (Webface) | FaceNet (Webface) | ArcFace |
| Baseline | 88.26± 0.19 | 83.95± 0.22 | 96.06± 0.16 | 97.41± 0.34 |
| AGENDA | 76.83± 0.57 | 74.51± 0.94 | 82.42± 0.45 | 95.09± 0.55 |
| PASS | 86.96± 0.46 | 81.44± 0.91 | 92.27± 0.72 | 96.55± 0.28 |
| FTC | 86.46± 0.17 | 81.61± 0.57 | 93.30± 0.70 | 96.41± 0.53 |
| GST | 89.57± 0.25 | 84.88± 0.18 | 96.59± 0.20 | 96.89± 0.38 |
| FSN | 90.05± 0.26 | 85.84± 0.34 | 96.77± 0.20 | 97.35± 0.33 |
| **FairCal (Ours)** | **90.58± 0.29** | **86.71± 0.25** | **96.90± 0.17** | **97.44± 0.34** |
| *Oracle (Ours)* | *89.74± 0.31* | *85.23± 0.18* | *97.28± 0.13* | *98.91± 0.12* |

Table 12: Global **accuracy** measured by the TPR at 0.1% FPR threshold.

| (↑) | RFW | | BFW | |
| --- | --- | --- | --- | --- |
| | FaceNet (VGGFace2) | FaceNet (Webface) | FaceNet (Webface) | ArcFace |
| Baseline | 18.42± 1.28 | 11.18± 3.45 | 33.61± 2.10 | 86.27± 1.09 |
| AGENDA | 8.32± 1.86 | 6.38± 0.78 | 15.95± 1.53 | 69.61± 2.40 |
| PASS | 13.67± 2.39 | 7.34± 1.74 | 17.21± 3.04 | 77.38± 1.51 |
| FTC | 6.86± 5.24 | 4.65± 2.10 | 13.60± 4.92 | 82.09± 1.11 |
| GST | 22.61± 1.52 | 17.34± 2.10 | 44.49± 1.13 | 86.13± 1.05 |
| FSN | 23.01± 2.00 | 17.33± 3.01 | **47.11± 1.23** | 86.19± 1.13 |
| **FairCal (Ours)** | **23.55± 1.82** | **20.64± 3.09** | 46.74± 1.49 | **86.28± 1.24** |
| *Oracle (Ours)* | *21.40± 3.54* | *16.71± 1.98* | *45.13± 1.45* | *86.41± 1.19* |

Table 13: Global **accuracy** measured by the TPR at 1% FPR threshold.

| (↑) | RFW | | BFW | |
| --- | --- | --- | --- | --- |
| | FaceNet (VGGFace2) | FaceNet (Webface) | FaceNet (Webface) | ArcFace |
| Baseline | 34.88± 3.27 | 26.04± 2.11 | 58.87± 0.92 | 90.11± 0.87 |
| AGENDA | 18.01± 1.44 | 14.98± 1.11 | 32.51± 1.24 | 79.67± 2.06 |
| PASS | 29.30± 2.24 | 20.93± 1.12 | 38.32± 5.48 | 85.26± 0.79 |
| FTC | 23.66± 6.58 | 18.40± 4.02 | 43.09± 5.70 | 88.24± 0.63 |
| GST | 40.72± 2.73 | 31.56± 1.18 | 66.71± 0.85 | 89.70± 0.85 |
| FSN | 40.21± 2.09 | 32.80± 1.03 | 68.92± 1.01 | 90.06± 0.84 |
| **FairCal (Ours)** | **41.88± 1.99** | **33.13± 1.67** | **69.21± 1.19** | **90.14± 0.86** |
| *Oracle (Ours)* | *41.83± 2.98* | *31.60± 1.08* | *67.56± 1.05* | *90.40± 0.91* |

Table 14: **Fairness-calibration** as measured by the mean KS across sensitive subgroups.

| (↓) | RFW | | BFW | |
| --- | --- | --- | --- | --- |
| | FaceNet (VGGFace2) | FaceNet (Webface) | FaceNet (Webface) | ArcFace |
| Baseline | 6.37± 0.35 | 5.55± 0.14 | 6.77± 0.57 | 2.57± 0.43 |
| AGENDA | 7.71± 0.27 | 5.71± 0.28 | 13.2± 1.04 | 5.14± 0.40 |
| PASS | 8.09± 1.16 | 7.65± 1.08 | 13.1± 2.90 | 3.69± 0.23 |
| FTC | 5.69± 0.14 | 4.73± 0.53 | 6.64± 0.41 | 2.95± 0.45 |
| GST | 2.34± 0.05 | 3.24± 0.40 | 3.09± 0.22 | 3.34± 0.28 |
| FSN | 1.43± 0.28 | 2.49± 0.46 | **2.76± 0.21** | 2.65± 0.43 |
| **FairCal (Ours)** | **1.37± 0.17** | **1.75± 0.26** | 3.09± 0.37 | **2.49± 0.43** |
| *Oracle (Ours)* | *1.18± 0.05* | *1.35± 0.09* | *2.23± 0.14* | *1.41± 0.33* |

Table 15: Bias in **fairness-calibration** as measured by the deviations of KS across subgroups in terms of AAD (Average Absolute Deviation).

| ($\downarrow$) | RFW | | BFW | |
|---|---|---|---|---|
| | FaceNet (VGGFace2) | FaceNet (Webface) | FaceNet (Webface) | ArcFace |
| Baseline | 2.89± 0.29 | 2.48± 0.36 | 3.63± 0.63 | 1.39± 0.28 |
| AGENDA | 3.11± 0.25 | 2.37± 0.33 | 6.37± 0.62 | 2.48± 0.50 |
| PASS | 2.40± 0.76 | 3.36± 0.87 | 5.25± 1.39 | 2.01± 0.20 |
| FTC | 2.32± 0.28 | 1.93± 0.35 | 2.80± 0.55 | 1.48± 0.31 |
| GST | 0.82± 0.22 | 1.21± 0.41 | 1.45± 0.27 | 1.81± 0.43 |
| FSN | 0.35± 0.15 | 0.84± 0.38 | 1.38± 0.27 | 1.45± 0.31 |
| **FairCal (Ours)** | **0.28± 0.12** | **0.41± 0.19** | **1.34± 0.24** | **1.30± 0.26** |
| *Oracle (Ours)* | *0.28± 0.08* | *0.38± 0.20* | *1.15± 0.24* | *0.59± 0.18* |

Table 16: Bias in **fairness-calibration** as measured by the deviations of KS across subgroups in terms of MAD (Maximum Absolute Deviation).

| ($\downarrow$) | RFW | | BFW | |
|---|---|---|---|---|
| | FaceNet (VGGFace2) | FaceNet (Webface) | FaceNet (Webface) | ArcFace |
| Baseline | 5.73± 0.63 | 4.97± 0.72 | 5.96± 1.05 | 2.94± 0.99 |
| AGENDA | 6.09± 0.65 | 4.28± 0.38 | 12.9± 0.47 | 5.92± 1.86 |
| PASS | 4.10± 1.11 | 5.34± 1.43 | 9.58± 1.81 | 4.24± 0.93 |
| FTC | 4.51± 0.64 | 3.86± 0.70 | 5.61± 0.66 | 3.03± 0.88 |
| GST | 1.58± 0.39 | 1.93± 0.56 | 2.80± 0.64 | 4.21± 1.38 |
| FSN | 0.57± 0.21 | 1.19± 0.38 | 2.67± 0.32 | 3.23± 0.99 |
| **FairCal (Ours)** | **0.50± 0.15** | **0.64± 0.28** | **2.48± 0.41** | **2.68± 1.07** |
| *Oracle (Ours)* | *0.53± 0.18* | *0.66± 0.28* | *2.63± 0.60* | *1.30± 0.29* |

Table 17: Bias in **fairness-calibration** as measured by the deviations of KS across subgroups in terms of STD (Standard Deviation).

| ($\downarrow$) | RFW | | BFW | |
|---|---|---|---|---|
| | FaceNet (VGGFace2) | FaceNet (Webface) | FaceNet (Webface) | ArcFace |
| Baseline | 3.77± 0.33 | 2.91± 0.41 | 4.03± 0.70 | 1.63± 0.40 |
| AGENDA | 3.86± 0.24 | 2.85± 0.33 | 7.55± 0.60 | 3.04± 0.65 |
| PASS | 2.83± 0.83 | 3.85± 0.91 | 6.12± 1.54 | 2.37± 0.30 |
| FTC | 2.95± 0.32 | 2.28± 0.43 | 3.27± 0.46 | 1.74± 0.42 |
| GST | 0.98± 0.26 | 1.34± 0.40 | 1.65± 0.26 | 2.19± 0.55 |
| FSN | 0.40± 0.15 | 0.91± 0.36 | 1.60± 0.23 | 1.71± 0.41 |
| **FairCal (Ours)** | **0.34± 0.12** | **0.45± 0.20** | **1.55± 0.24** | **1.52± 0.37** |
| *Oracle (Ours)* | *0.33± 0.10* | *0.43± 0.20* | *1.40± 0.27* | *0.69± 0.18* |

Table 18: **Predictive equality:** Each block of rows represents a choice of global FPR: 0.1% and 1%. For a fixed a global FPR, compare the deviations in subgroup FPRs in terms of AAD (Average Absolute Deviation). We report the average and standard deviation error across the 5 folds.

| | | RFW | | BFW | |
|---|---|---|---|---|---|
| | ($\downarrow$) | FaceNet (VGGFace2) | FaceNet (Webface) | FaceNet (Webface) | ArcFace |
| 0.1% FPR | Baseline | 0.10± 0.02 | 0.14± 0.03 | 0.29± 0.04 | 0.12± 0.03 |
| | AGENDA | 0.11± 0.04 | 0.12± 0.03 | 0.14± 0.04 | **0.09± 0.03** |
| | PASS | 0.11± 0.02 | 0.11± 0.02 | 0.36± 0.03 | 0.12± 0.03 |
| | FTC | 0.10± 0.02 | 0.12± 0.04 | 0.24± 0.02 | **0.09± 0.02** |
| | GST | 0.13± 0.02 | **0.09± 0.02** | 0.13± 0.03 | 0.10± 0.03 |
| | FSN | 0.10± 0.05 | 0.11± 0.04 | **0.09± 0.03** | 0.11± 0.02 |
| | **FairCal (Ours)** | **0.09± 0.03** | **0.09± 0.03** | **0.09± 0.02** | 0.11± 0.03 |
| | *Oracle (Ours)* | *0.11± 0.05* | *0.11± 0.03* | *0.12± 0.03* | *0.12± 0.04* |
| 1% FPR | Baseline | 0.68± 0.06 | 0.67± 0.15 | 2.42± 0.14 | 0.72± 0.19 |
| | AGENDA | 0.73± 0.11 | 0.73± 0.08 | 1.21± 0.27 | 0.65± 0.13 |
| | PASS | 0.89± 0.10 | 0.68± 0.19 | 3.30± 0.25 | 0.72± 0.13 |
| | FTC | 0.60± 0.11 | 0.54± 0.12 | 1.94± 0.22 | 0.54± 0.09 |
| | GST | 0.52± 0.12 | 0.30± 0.03 | 1.05± 0.15 | **0.44± 0.13** |
| | FSN | 0.37± 0.12 | 0.35± 0.16 | 0.87± 0.11 | 0.55± 0.11 |
| | **FairCal (Ours)** | **0.28± 0.11** | **0.29± 0.10** | **0.80± 0.10** | 0.63± 0.15 |
| | *Oracle (Ours)* | *0.40± 0.09* | *0.41± 0.10* | *0.77± 0.17* | *0.83± 0.15* |

Table 19: **Predictive equality:** Each block of rows represents a choice of global FPR: 0.1% and 1%. For a fixed a global FPR, compare the deviations in subgroup FPRs in terms of MAD (Maximum Absolute Deviation). We report the average and standard deviation error across the 5 folds.

| | | RFW | | BFW | |
|---|---|---|---|---|---|
| | ($\downarrow$) | FaceNet (VGGFace2) | FaceNet (Webface) | FaceNet (Webface) | ArcFace |
| 0.1% FPR | Baseline | 0.15± 0.05 | 0.26± 0.09 | 1.00± 0.28 | 0.30± 0.08 |
| | AGENDA | 0.20± 0.10 | 0.23± 0.07 | 0.40± 0.16 | 0.23± 0.10 |
| | PASS | 0.18± 0.04 | 0.18± 0.03 | 1.21± 0.26 | 0.29± 0.11 |
| | FTC | 0.15± 0.03 | 0.23± 0.08 | 0.74± 0.22 | **0.20± 0.03** |
| | GST | 0.24± 0.06 | **0.16± 0.04** | 0.35± 0.16 | 0.24± 0.06 |
| | FSN | 0.18± 0.10 | 0.23± 0.07 | **0.20± 0.06** | 0.28± 0.08 |
| | **FairCal (Ours)** | **0.14± 0.04** | **0.16± 0.06** | **0.20± 0.04** | 0.31± 0.10 |
| | *Oracle (Ours)* | *0.19± 0.10* | *0.20± 0.07* | *0.25± 0.06* | *0.27± 0.09* |
| 1% FPR | Baseline | 1.02± 0.01 | 1.23± 0.30 | 7.48± 1.75 | 1.51± 0.44 |
| | AGENDA | 1.14± 0.22 | 1.08± 0.10 | 3.09± 1.06 | 1.78± 0.76 |
| | PASS | 1.52± 0.37 | 0.99± 0.08 | 10.1± 2.31 | 2.00± 0.64 |
| | FTC | 0.91± 0.08 | 1.05± 0.17 | 5.74± 1.73 | **1.04± 0.15** |
| | GST | 0.92± 0.25 | 0.57± 0.06 | 3.01± 1.04 | 1.13± 0.39 |
| | FSN | 0.68± 0.23 | 0.61± 0.25 | 2.19± 0.58 | 1.27± 0.35 |
| | **FairCal (Ours)** | **0.46± 0.16** | **0.57± 0.23** | **1.79± 0.54** | 1.46± 0.29 |
| | *Oracle (Ours)* | *0.69± 0.19* | *0.74± 0.23* | *1.71± 0.59* | *2.08± 0.57* |

Table 20: **Predictive equality:** Each block of rows represents a choice of global FPR: 0.1% and 1%. For a fixed a global FPR, compare the deviations in subgroup FPRs in terms of STD (Standard Deviation). We report the average and standard deviation error across the 5 folds.

| | | RFW | | BFW | |
|---|---|---|---|---|---|
| | ($\downarrow$) | FaceNet (VGGFace2) | FaceNet (Webface) | FaceNet (Webface) | ArcFace |
| **0.1% FPR** | Baseline | 0.10± 0.03 | 0.16± 0.04 | 0.40± 0.09 | 0.15± 0.04 |
| | AGENDA | 0.13± 0.05 | 0.14± 0.04 | 0.18± 0.05 | **0.11± 0.04** |
| | PASS | 0.12± 0.03 | 0.12± 0.02 | 0.49± 0.08 | 0.14± 0.04 |
| | FTC | 0.11± 0.02 | 0.14± 0.05 | 0.32± 0.05 | **0.11± 0.02** |
| | GST | 0.15± 0.03 | **0.10± 0.03** | 0.16± 0.05 | 0.12± 0.03 |
| | FSN | 0.11± 0.06 | 0.13± 0.04 | **0.11± 0.03** | 0.14± 0.03 |
| | **FairCal (Ours)** | **0.10± 0.03** | **0.10± 0.03** | **0.11± 0.03** | 0.15± 0.03 |
| | *Oracle (Ours)* | *0.12± 0.05* | *0.13± 0.03* | *0.15± 0.03* | *0.14± 0.04* |
| **1% FPR** | Baseline | 0.74± 0.04 | 0.79± 0.18 | 3.22± 0.44 | 0.85± 0.20 |
| | AGENDA | 0.81± 0.11 | 0.78± 0.06 | 1.51± 0.33 | 0.84± 0.23 |
| | PASS | 1.01± 0.16 | 0.73± 0.15 | 4.34± 0.53 | 0.93± 0.19 |
| | FTC | 0.66± 0.09 | 0.66± 0.12 | 2.57± 0.45 | 0.61± 0.08 |
| | GST | 0.60± 0.14 | 0.37± 0.03 | 1.38± 0.29 | **0.56± 0.18** |
| | FSN | 0.46± 0.14 | 0.40± 0.17 | 1.05± 0.18 | 0.68± 0.14 |
| | **FairCal (Ours)** | **0.32± 0.12** | **0.35± 0.13** | **0.95± 0.16** | 0.78± 0.15 |
| | *Oracle (Ours)* | *0.45± 0.11* | *0.48± 0.12* | *0.91± 0.22* | *1.07± 0.18* |

Table 21: **Equal opportunity:** Each block of rows represents a choice of global FNR: 0.1% and 1%. For a fixed a global FNR, compare the deviations in subgroup FNRs in terms of AAD (Average Absolute Deviation). We report the average and standard deviation error across the 5 folds.

| | | RFW | | BFW | |
|---|---|---|---|---|---|
| | ($\downarrow$) | FaceNet (VGGFace2) | FaceNet (Webface) | FaceNet (Webface) | ArcFace |
| **0.1% FPR** | Baseline | 0.09± 0.01 | 0.10± 0.02 | 0.09± 0.03 | 0.11± 0.02 |
| | AGENDA | 0.11± 0.04 | 0.10± 0.02 | 0.14± 0.01 | 0.09± 0.02 |
| | PASS | **0.09± 0.03** | 0.11± 0.03 | 0.10± 0.02 | 0.10± 0.02 |
| | FTC | **0.09± 0.01** | **0.08± 0.03** | **0.04± 0.02** | **0.06± 0.01** |
| | GST | **0.09± 0.02** | 0.12± 0.02 | 0.10± 0.03 | 0.12± 0.01 |
| | FSN | **0.09± 0.02** | 0.09± 0.02 | 0.07± 0.02 | 0.12± 0.01 |
| | **FairCal (Ours)** | 0.10± 0.02 | 0.11± 0.02 | 0.10± 0.02 | 0.09± 0.02 |
| | *Oracle (Ours)* | *0.11± 0.02* | *0.12± 0.02* | *0.09± 0.02* | *0.11± 0.02* |
| **1% FPR** | Baseline | 0.60± 0.17 | 0.45± 0.09 | 0.39± 0.05 | 0.75± 0.16 |
| | AGENDA | 0.99± 0.24 | 0.67± 0.17 | 0.90± 0.09 | 0.72± 0.19 |
| | PASS | 0.77± 0.10 | 0.83± 0.14 | 0.72± 0.13 | 0.71± 0.16 |
| | FTC | 0.48± 0.06 | **0.32± 0.12** | **0.30± 0.07** | **0.49± 0.14** |
| | GST | 0.39± 0.05 | 0.54± 0.16 | 0.49± 0.12 | 0.83± 0.19 |
| | FSN | **0.28± 0.06** | 0.40± 0.19 | 0.41± 0.10 | 0.77± 0.17 |
| | **FairCal (Ours)** | 0.30± 0.14 | 0.39± 0.12 | 0.32± 0.10 | 0.65± 0.11 |
| | *Oracle (Ours)* | *0.38± 0.15* | *0.56± 0.11* | *0.37± 0.09* | *0.50± 0.10* |

Table 22: **Equal opportunity:** Each block of rows represents a choice of global FNR: 0.1% and 1%. For a fixed a global FNR, compare the deviations in subgroup FNRs in terms of MAD (Maximum Absolute Deviation). We report the average and standard deviation error across the 5 folds.

| | | RFW | | BFW | |
|---|---|---|---|---|---|
| | (↓) | FaceNet (VGGFace2) | FaceNet (Webface) | FaceNet (Webface) | ArcFace |
| 0.1% FPR | Baseline | 0.13± 0.02 | 0.16± 0.08 | 0.23± 0.11 | 0.31± 0.07 |
| | AGENDA | 0.22± 0.08 | 0.14± 0.08 | 0.34± 0.05 | 0.24± 0.09 |
| | PASS | 0.14± 0.06 | 0.20± 0.08 | 0.33± 0.12 | 0.31± 0.14 |
| | FTC | **0.11± 0.02** | **0.14± 0.07** | **0.09± 0.03** | **0.14± 0.04** |
| | GST | 0.13± 0.06 | 0.21± 0.07 | 0.26± 0.12 | 0.37± 0.05 |
| | FSN | 0.13± 0.06 | **0.14± 0.06** | 0.22± 0.11 | 0.33± 0.06 |
| | **FairCal (Ours)** | 0.14± 0.06 | 0.17± 0.09 | 0.27± 0.09 | 0.17± 0.05 |
| | *Oracle (Ours)* | *0.18± 0.07* | *0.21± 0.08* | *0.24± 0.08* | *0.32± 0.15* |
| 1% FPR | Baseline | 0.96± 0.21 | 0.81± 0.14 | 0.84± 0.14 | 1.85± 0.66 |
| | AGENDA | 1.97± 0.48 | 1.33± 0.35 | 2.39± 0.74 | 1.54± 0.42 |
| | PASS | 1.06± 0.20 | 1.64± 0.28 | 1.79± 0.71 | 1.76± 0.74 |
| | FTC | 0.83± 0.21 | **0.58± 0.24** | **0.62± 0.11** | **1.12± 0.29** |
| | GST | 0.60± 0.07 | 0.96± 0.35 | 1.02± 0.39 | 2.27± 0.76 |
| | FSN | **0.47± 0.15** | 0.78± 0.38 | 0.92± 0.28 | 1.91± 0.67 |
| | **FairCal (Ours)** | 0.51± 0.25 | 0.72± 0.20 | 0.74± 0.18 | 1.48± 0.36 |
| | *Oracle (Ours)* | *0.61± 0.15* | *1.06± 0.18* | *0.77± 0.19* | *1.11± 0.27* |

Table 23: **Equal opportunity:** Each block of rows represents a choice of global FNR: 0.1% and 1%. For a fixed a global FNR, compare the deviations in subgroup FNRs in terms of STD (Standard Deviation). We report the average and standard deviation error across the 5 folds.

| | | RFW | | BFW | |
|---|---|---|---|---|---|
| | (↓) | FaceNet (VGGFace2) | FaceNet (Webface) | FaceNet (Webface) | ArcFace |
| 0.1% FPR | Baseline | 0.10± 0.01 | 0.11± 0.03 | 0.11± 0.04 | 0.14± 0.02 |
| | AGENDA | 0.13± 0.04 | 0.11± 0.03 | 0.16± 0.02 | 0.12± 0.03 |
| | PASS | 0.10± 0.03 | 0.12± 0.04 | 0.14± 0.04 | 0.14± 0.04 |
| | FTC | **0.09± 0.01** | **0.10± 0.03** | **0.05± 0.02** | **0.07± 0.02** |
| | GST | 0.10± 0.03 | 0.13± 0.03 | 0.12± 0.04 | 0.15± 0.02 |
| | FSN | **0.09± 0.03** | 0.10± 0.03 | 0.10± 0.04 | 0.15± 0.02 |
| | **FairCal (Ours)** | 0.10± 0.02 | 0.12± 0.03 | 0.13± 0.03 | 0.10± 0.02 |
| | *Oracle (Ours)* | *0.12± 0.03* | *0.13± 0.03* | *0.11± 0.03* | *0.14± 0.04* |
| 1% FPR | Baseline | 0.67± 0.15 | 0.53± 0.09 | 0.47± 0.06 | 0.93± 0.23 |
| | AGENDA | 1.16± 0.27 | 0.81± 0.19 | 1.15± 0.17 | 0.84± 0.22 |
| | PASS | 0.81± 0.10 | 0.99± 0.15 | 0.89± 0.16 | 0.90± 0.24 |
| | FTC | 0.56± 0.10 | **0.38± 0.15** | **0.34± 0.06** | **0.60± 0.16** |
| | GST | 0.44± 0.04 | 0.62± 0.18 | 0.57± 0.14 | 1.07± 0.26 |
| | FSN | **0.32± 0.09** | 0.48± 0.23 | 0.49± 0.12 | 0.96± 0.23 |
| | **FairCal (Ours)** | 0.34± 0.17 | 0.48± 0.14 | 0.40± 0.10 | 0.80± 0.13 |
| | *Oracle (Ours)* | *0.42± 0.14* | *0.67± 0.11* | *0.44± 0.10* | *0.60± 0.12* |

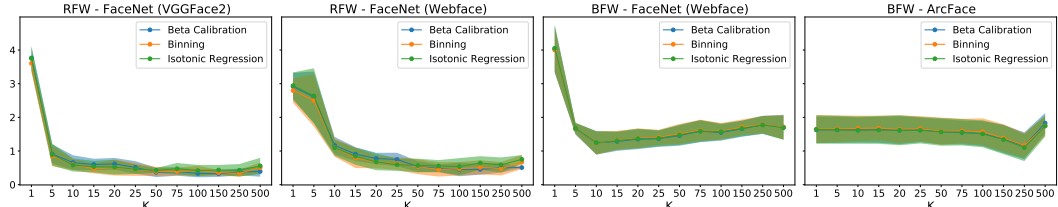

Figure 7: Bias in **fairness-calibration** as measured by the STD (Standard Deviation) in the KS across the sensitive subgroups for different values of $K$ and different choices of post-hoc calibration methods. Shaded regions refer to the standard error across the 5 different folds in the datasets.

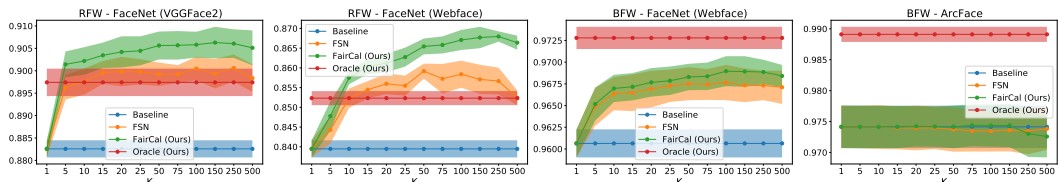

Figure 8: Global **accuracy** measured by the AUROC for different values of $K$ for Baseline, FSN Terhörst et al. (2020b), FairCal, and Oracle methods. Shaded regions refer to the standard error across the 5 different folds in the datasets.

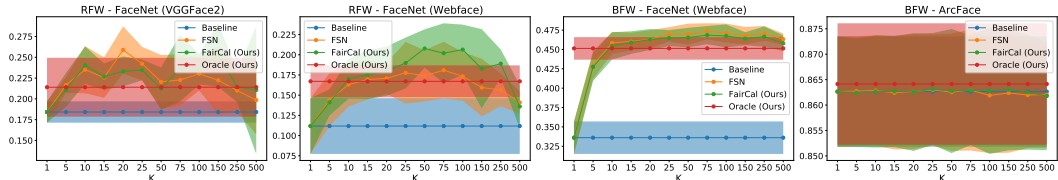

Figure 9: Global **accuracy** measure by the TPR at different a global 0.1% FPR for different values of $K$ for Baseline, FSN Terhörst et al. (2020b), FairCal, and Oracle methods. Shaded regions refer to the standard error across the 5 different folds in the datasets.

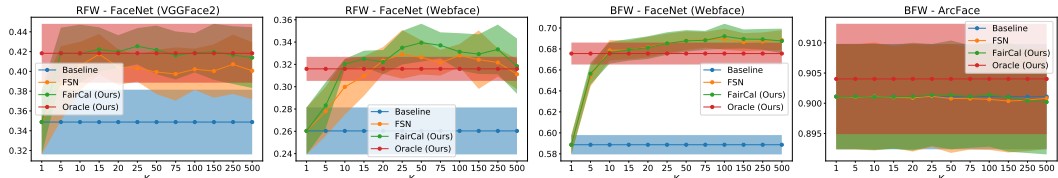

Figure 10: Global **accuracy** measure by the TPR at different a global 1% FPR for different values of $K$ for Baseline, FSN Terhörst et al. (2020b), FairCal, and Oracle methods. Shaded regions refer to the standard error across the 5 different folds in the datasets.

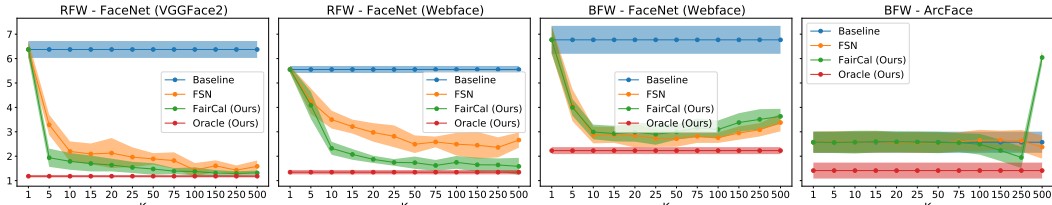

Figure 11: Comparison of **fairness-calibration** as measured by the subgroup mean of the KS across the sensitive subgroups for different values of $K$ for Baseline, FSN Terhörst et al. (2020b), FairCal, and Oracle methods. Shaded regions refer to the standard error across the 5 different folds in the datasets.

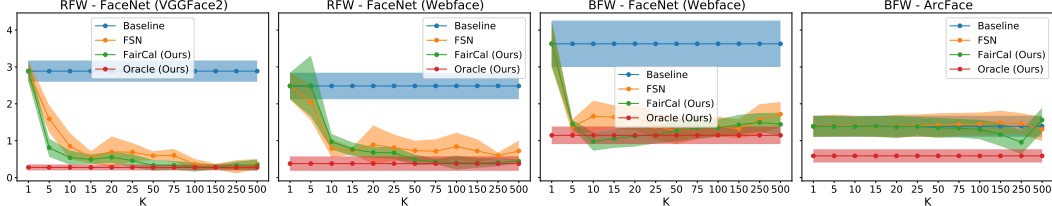

Figure 12: Bias in **fairness-calibration** as measured by the AAD (Average Absolute Deviation) in the KS across the sensitive subgroups for different values of $K$ for Baseline, FSN Terhörst et al. (2020b), FairCal, and Oracle methods. Shaded regions refer to the standard error across the 5 different folds in the datasets.

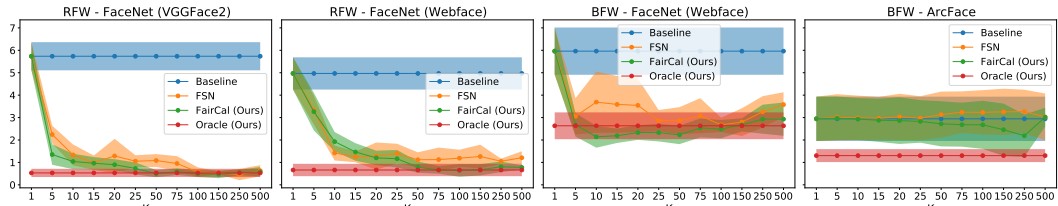

Figure 13: Bias in **fairness-calibration** as measured by the MAD (Maximum Absolute Deviation) in the KS across the sensitive subgroups for different values of $K$ for Baseline, FSN Terhörst et al. (2020b), FairCal, and Oracle methods. Shaded regions refer to the standard error across the 5 different folds in the datasets.

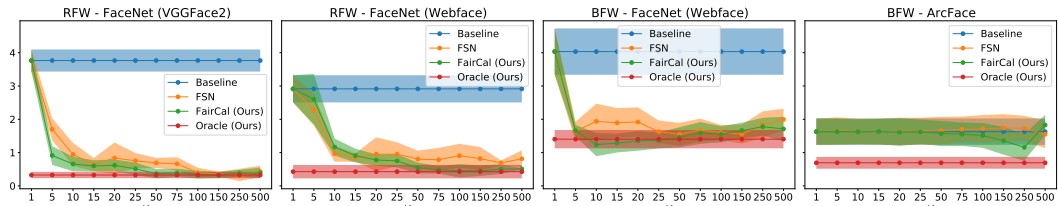

Figure 14: Bias in **fairness-calibration** as measured by the STD (Standard Deviation) in the KS across the sensitive subgroups for different values of $K$ for Baseline, FSN Terhörst et al. (2020b), FairCal, and Oracle methods. Shaded regions refer to the standard error across the 5 different folds in the datasets.

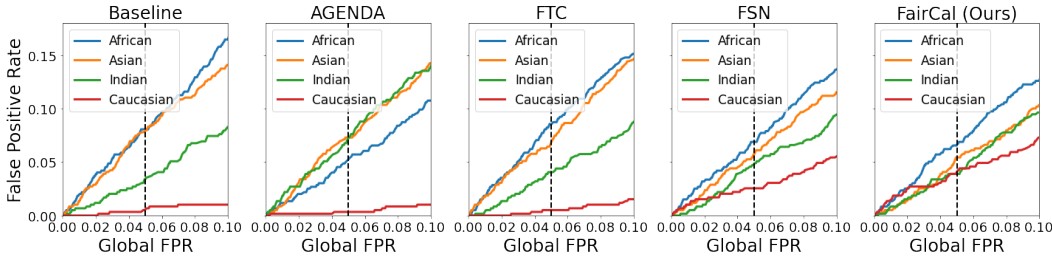

Figure 15: (Lines closer together is better for fairness, best viewed in colour) Illustration of improved fairness / reduction in bias, as measured by the FPRs evaluated on intra-ethnicity pairs on the RFW dataset with the FaceNet (VGGFace2) feature model.

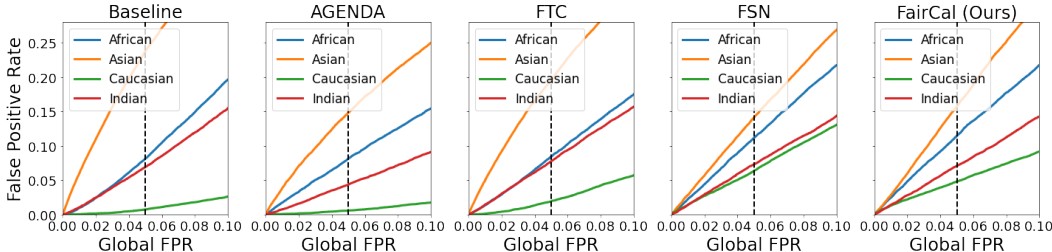

Figure 16: (Lines closer together is better for fairness, best viewed in colour) Illustration of improved fairness / reduction in bias, as measured by the FPRs evaluated on intra-ethnicity pairs on the BFW dataset with the FaceNet (Webface) feature model.

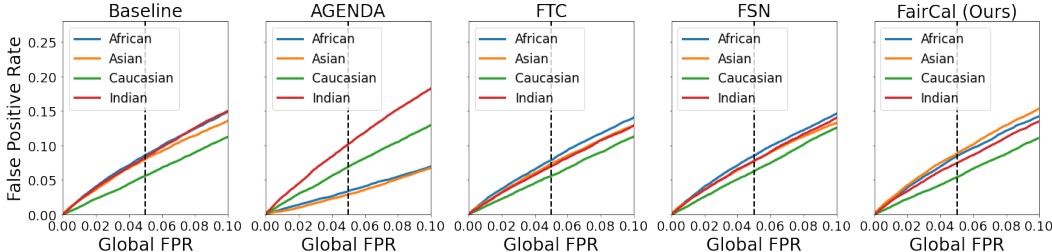

Figure 17: (Lines closer together is better for fairness, best viewed in colour) Illustration of improved fairness / reduction in bias, as measured by the FPRs evaluated on intra-ethnicity pairs on the BFW dataset with the ArcFace feature model.

