# OpenReview forum: "FairCal: Fairness Calibration for Face Verification"
_ICLR.cc/2022/Conference — ICLR 2022 Poster_

### Official Review · Reviewer_JNHP · 2021-11-01

**Correctness:** 3
**Technical Novelty And Significance:** 2
**Empirical Novelty And Significance:** Not applicable
**Recommendation:** 6
**Confidence:** 5

**Main Review:**

Strength
+ Simple approach for post-hoc bias mitigation.
+ Improvement over SOTA Arcface
+ Extensive evaluation using different fairness metrics.

Weaknesses
- (Major) No comparison with group-specific thresholding proposed by Robinson et al (CVPRW 2020).
Robinson et al  (CVPRW 2020) propose a similar method and report results on BFW dataset. Additionally, they also report improvement in face verification accuracy. This makes me question the novelty of this work. The authors are advised to mention the key differences and add comparisons with this work.

- Missing analysis of 'Baseline + Calibration (2nd plot in Fig. 2)' in Table 2,3,4.

- Results (Table 7).
From this table, can we infer that FairCal does not achieve Equal Opportunity (compared to FTC, FSN)?

- Missing details about building calibration set.  How are samples in the calibration set chosen?
- Missing notational details (For ex: in Section F1 and F2, what is P^{cal} )


- (Minor) Missing citations and comparison
'Group Adaptive Classifier' by Gong et al (CVPR 2021)  and 'Protected Attribute Suppression System'  by Dhar et al (ICCV 2021) recently proposed debiasing methods. While Gong et al (CVPR 2021) present an end-to-end method, Dhar et al (ICCV 2021) present a post-hoc de-biasing approach, that is comparable to FairCal.

- (Minor) IJB-C is one of the most widely used datasets for evaluating face recognition algorithms, which contains gender and skin tone labels. So, it is recommended that the authors add results related to skin tone  and gender bias on IJBC.

-  (Suggestion, not weakness) The authors are suggested to add plots similar to Fig 1 for Arcface on BFW. Also, it may help to add some more intuition behind Eq 3.


**Summary Of The Paper:**

The authors propose a simple cluster-conditional post-hoc strategy to calibrate the similarity scores obtained using a pre-trained network. In essence, this work proposes to use a cluster-specific (or group-specific in case of Oracle) calibration function (\mu), which results in improving the scores pertaining to verification of a x1-x2 pair (where both x1 and x2 belong to the same demographic group) mapping function. This is similar to choosing group-specific threshold demonstrated in the BFW paper (Robinson et al, CVPRW 2020). Unlike existing methods such as AGENDA, FTC etc. FairCal does not require the protected attribute labels during training and testing.

**Summary Of The Review:**

The topic of this paper is currently very important in the CV/ML community. Although the novelty of this work is incremental,  the results justify the superiority of FairCal (and oracle). While the readability and thoroughness of this work is quite high, there are still some issues in this work. So, I vote for weak accept. I hope the authors address my concerns (especially about similarities with the BFW paper and add comparison to this paper).

---

> ### Author Response · Authors · 2021-11-16
> **Response JNHP (1/2)**
>
> We thank the reviewer for their time and effort in reviewing our paper. Please find below our responses to your questions, we are happy to provide any more clarifications necessary.
>
> > - (Major) No comparison with group-specific thresholding proposed by Robinson et al (CVPRW 2020). Robinson et al (CVPRW 2020) propose a similar method and report results on BFW dataset. Additionally, they also report improvement in face verification accuracy. This makes me question the novelty of this work. The authors are advised to mention the key differences and add comparisons with this work.
>
> Thank you for pointing out that Robinson et al (CVPRW 2020) which, besides proposing the dataset BFW, proposes a bias mitigation method that uses a group-specific threshold instead of a global threshold. We will refer to this method as GST (Group-Specific Thresholding).
>
> GST is very similar to the FSN method, and suffers from many limitations, some of which are shared with FSN. In particular,
> 1) GST depends on a global FPR chosen a priori. Our FairCal does not.
> 2) GST defines calibration sets based on the sensitive attribute, while FSN and our FairCal choose clusters in an unsupervised fashion.
> 3) GST (thus) requires knowing the sensitive attribute not only during training, but also at testing. As mentioned in the paper for our FairCal method, we "assume that we do not have access to the sensitive attributes, as this may not be feasible in practice due to a) privacy concerns, b) challenges in defining the sensitive attribute (e.g. ethnicity cannot be neatly divided into discrete categories), and c) laborious and expensive to collect."
>
> In short, GST is FSN with access to sensitive attributes, i.e. GST is to FSN like our Oracle method is to our FairCal.
>
> We have revised the paper mentioning these points, and include GST as an additional baseline comparison. Despite not using sensitive attributes during training and testing, our FairCal method consistently outperforms GST.
>
> > - Missing analysis of 'Baseline + Calibration (2nd plot in Fig. 2)' in Table 2,3,4.
>
> Thank you for this question. We emphasize that standard post-hoc calibration alone, which is precisely the case in 'Baseline + Calibration', does not reduce mitigation. In fact, when the calibration map $\mu$ is restricted to be monotone (a standard assumption (Platt et al., 1999, Zadrozny & Elkan, 2002)), the bias remains exactly the same. As we mention at the end of Section 3.2, "a binary classifier with a score threshold $s_{thr}$ can be obtained by thresholding the probabilistic classifier at $\mu(s_{thr})$." The intuition is that the calibration map changes the scale of the scores but their ordering remains the same. Since all the metrics displayed in Tables 2, 3, 4 result from thresholding, 'Baseline' and 'Baseline + Calibration' lead to the exact same results.
>
> > - Results (Table 7). From this table, can we infer that FairCal does not achieve Equal Opportunity (compared to FTC, FSN)?
>
> We wish to point out that none of the methods were designed to achieve Equal Opportunity in the first place. AGENDA, FTC were designed to learn less biased representations, FSN was designed to achieve Predictive Equality, while our FairCal method was designed to achieve Fairness-calibration.
>
> Table 7 shows that FTC is the method that more consistenly shows Equal Opportunity. However, this comes at significant drop in performance (see Table 2). As for FairCal and FSN, we argue that they are comparable in their effectiveness. For the BFW dataset, FairCal has better results for Equal Opportunity, while FSN does for RFW.
>
> > - Missing details about building calibration set. How are samples in the calibration set chosen?
>
> As mentioned in Section 5, "both datasets already include predefined pairs separated into five folds. The results we present are the product of leave-one-out cross-validation." To be more specific, one fold is used as the test set and the remaining folds form the calibration sets. The pairs in the folds are chosen such that there are no overlapping identities and the ratio between positive and negative pairs is constant across folds. These details have been included in the original papers that proposed the dataset, please check Wang et al. (2019a) (RFW) and Robinson et al. (2020).
>
> > - Missing notational details (For ex: in Section F1 and F2, what is $\mathcal{P}^{cal}$ )
>
> Thank you for pointing this out! We have reviewed both Section F1 and F2 (now sections G1 and G2) and added the missing details in the revised version of the paper, in particular the definition of $\mathcal{P}^{cal}$. Please let us know if you find any more missing details.

---

> > ### Author Response · Authors · 2021-11-16
> > **Response JNHP (2/2)**
> >
> > > - (Minor) Missing citations and comparison 'Group Adaptive Classifier' by Gong et al (CVPR 2021) and 'Protected Attribute Suppression System' by Dhar et al (ICCV 2021) recently proposed debiasing methods. While Gong et al (CVPR 2021) present an end-to-end method, Dhar et al (ICCV 2021) present a post-hoc de-biasing approach, that is comparable to FairCal.
> >
> > Thank you for pointing out the missing references. We have added them and included a comparison to Dhar et al (ICCV 2021) which proposes PASS, in the revised version of our paper. Our FairCal method outperforms PASS.
> >
> > To note as well, as the reviewer points out, Gong et al (CVPR 2021) propose to mitigate bias in an end-to-end fashion by learning less biased representations, in contrast to our post-processing approach with FairCal. They propose a Group Adaptive Classifier (GAC) that leverages adaptive convolution kernels and attention mechanisms. In addition, the models are trained on a new de-biasing loss function that requires knowing the sensitive attribute (unlike FairCal which does not need access to sensitive attributes).
> >
> > Dhar et al (ICCV 2021) propose PASS, an extension of AGENDA (Dhar et al. , 2020), where a shallow network is trained to learn a new debiased embedding. This introduces several hyper-parameters and requires knowing the sensitive attribute. In contrast, our FairCal approach is a simple statiscal approach relying only on the standard algorithms ($K$-means and post-hoc calibration), and more importantly it does not require knowing the sensitive attribute. Moreover, like AGENDA, PASS led to a decrease in performance, while FairCal improves performance (see Table 2).
> >
> > > - (Minor) IJB-C is one of the most widely used datasets for evaluating face recognition algorithms, which contains gender and skin tone labels. So, it is recommended that the authors add results related to skin tone and gender bias on IJBC.
> >
> > Thank you for the suggestion. We have taken steps to add the results, but requesting the dataset and downloading it took 48 hours. We will try to add the results by the end of the rebuttal period. If we are unable to do so, we will certainly include them in the camera-ready version if the paper is accepted.
> >
> > > - (Suggestion, not weakness) The authors are suggested to add plots similar to Fig 1 for Arcface on BFW. Also, it may help to add some more intuition behind Eq 3.
> >
> > We have added in the Appendix similar plots to Figure 1 for FaceNet (VGGFace2) on RFW, FaceNet (Webface) on BFW and Arcface on BFW.
> >
> > Eq 3 is the weighting based on the calibration set size when a pair belongs to two different clusters. If one calibration set has more pairs than the other, we give it more weight. In the extreme case, that one calibration set has 900 pairs and the other has only 100 pairs, we expect the calibration of the 900 pairs to be more accurate than if just using 100 pairs. The weighting accounts for that giving weights 0.9 and 0.1, respectively.
> >
> > > I vote for weak accept. I hope the authors address my concerns (especially about similarities with the BFW paper and add comparison to this paper).
> >
> > We hope to have addressed all your major comments, in particular with the additional comparison to the BFW paper. We are happy to provide any further clarifications. In light of our response, we hope the reviewer increases their rating.

---

> > > ### Comment · Reviewer_JNHP · 2021-11-22
> > > **Response to authors**
> > >
> > > I thank the authors for their detailed response.
> > >
> > > There are a few issues with the comparisons to PASS. According to the original PASS paper (Dhar et al, ICCV 2021), PASS reduces bias compared to the original network. However, in tables 4, 5 in the manuscript, I do not see this trend. This is also true for  Tables 5, 6, 7, 11,12, 13 in supplementary material. This makes me question baseline implementation.
> > >
> > > Moreover, the  manuscript also currently does not contain the results on the IJBC dataset (which is more widely used than RFW).
> > >
> > > Citing these reasons, I would like to keep my rating unchanged.

---

> > > > ### Author Response · Authors · 2021-11-27
> > > > **Further clarification on comparison to PASS**
> > > >
> > > > > There are a few issues with the comparisons to PASS. According to the original PASS paper (Dhar et al, ICCV 2021), PASS reduces bias compared to the original network. However, in tables 4, 5 in the manuscript, I do not see this trend. This is also true for Tables 5, 6, 7, 11,12, 13 in supplementary material. This makes me question baseline implementation.
> > > >
> > > > Thank you for this question as it gives us the opportunity to elaborate on our empirical evaluation procedure. We follow the recommendations of the report by the National Institute of Standards and Technology (NIST) [1]. In it, the following is stated:
> > > >
> > > > *"A crucial point in reasoning about [accuracy] differentials is that the vast majority of biometric systems are configured with a fixed threshold against which all comparisons are made (i.e., the threshold is not tailored to cameras, environmental conditions or, particularly, demographics). Most academic studies ignore this point (even in demographics e.g., [2]) by reporting false negative rates at fixed false positive rates rather than at fixed thresholds, thereby hiding excursions in false positive rates and misstating false negative rates."*
> > > >
> > > > - The PASS paper suffers from this. Bias is measured in PASS by comparing the True Positive Rate (TPR) for each subgroup at a given False Positive Rate (FPR). In contrast, and as recommended in [1], we report false positive rates (Predictive Equality, see Table 4) and false negative rates (Equal Opportunity, see Table 7) separately.
> > > >
> > > > - In addition, PASS reports bias by comparing only two subgroups (Male versus Female, Dark versus Light skin tone). In contrast, our results include 4 subgroups (RFW), 8 subgroups (BFW), 12 subgroups (IJB-C).
> > > >
> > > > - Moreover, our results on IJB-C are in agreement with our results on RFW and BFW, i.e. we see a consistent trendline across multiple models and datasets.
> > > >
> > > > Regarding our implementation of PASS:
> > > > 1. We followed the suggestions of the original paper for all the hyperparameters, and adopted the same shallow network architectures.
> > > > 2. PASS only reports a "trendline" for two model and ONE dataset. We report numbers for multiple models and multiple datasets. If anything, we are the ones showing a trendline.
> > > > 3. If the reviewer still thinks our implementation must be wrong, it would be great if the reviewer could please check our code and tell us where the error is. The code from our first submission can be found in the supplemental material and updated code with the code for PASS can be found here: https://anonymous.4open.science/r/faircal/.
> > > >
> > > > Finally, we would like to point out that the PASS paper was made available on arXiv on Aug 9, 2021, i.e. after the ICLR deadline of June 5, 2021 to compare with recent work. The official implementation of PASS (https://github.com/Prithviraj7/PASS) was not available until Nov 16, 2021, i.e. after we posted the revision of our paper. Nevertheless, we implemented PASS, reported our results on it, and made its code available. We hope the reviewer recognizes all these efforts made by us to be transparent and forthcoming about our method, as well as its superiority over PASS in all aspects.
> > > >
> > > > > Moreover, the manuscript also currently does not contain the results on the IJBC dataset (which is more widely used than RFW).
> > > >
> > > > We were able to obtain results for two of our models on the IJB-C dataset. The samples chosen follow the same guidelines as in the RFW and BFW dataset. The results are below and they are in agreement with the results obtained on RFW and BFW.
> > > >
> > > > [1] Grother, P. , Ngan, M. and Hanaoka, K. (2019), Face Recognition Vendor Test Part 3: Demographic Effects, NIST Interagency/Internal Report (NISTIR), National Institute of Standards and Technology, Gaithersburg, MD, [online], https://doi.org/10.6028/NIST.IR.8280
> > > >
> > > > [2] H El Khiyari and Wechsler H. Face verification subject to varying (age, ethnicity, and gender) demographics using deep learning. Journal of Biometrics and Biostatistics, 7:323, 11 2016.

---

> > > > > ### Author Response · Authors · 2021-11-27
> > > > > **(Partial) Results on IJB-C**
> > > > >
> > > > > Below are the results on IJB-C we were able to obtain. We will add the results for the FTC method and for the different methods for the ArcFace model to the camera-ready version if the paper is accepted. The FTC method requires a small batch size and works directly on the pairs (unlike AGENDA and PASS) and therefore given the short rebuttal period ending in two days there was not enough time to compute it.
> > > > >
> > > > > Global accuracy measured by the AUROC and the TPR at different FPR thresholds.
> > > > >
> > > > > | | FaceNet (VGGFace2) |          |           | FaceNet (Webface) | | |
> > > > > |-----------------|:------------------:|----------|:---------:|:-----------------:|:---------:|-----------|
> > > > > |                 |        AUROC       | 0.1% FPR |   1% FPR  | AUROC             |  0.1% FPR | 1% FPR    |
> > > > > | Baseline        |        92.72       | 43.53    |   62.80   | 89.23             |   19.50   | 50.10     |
> > > > > | AGENDA          |        89.86       | 44.78    |   61.65   | 81.33             |   32.90   | 46.71     |
> > > > > | PASS            |        86.68       | 24.06    |   39.46   | 81.54             |    9.89   | 28.66     |
> > > > > | GST             |        92.78       | 47.65    |   66.34   | 89.13             |   25.06   | 50.57     |
> > > > > | FSN             |        92.48       | **53.40** |   68.29   | 88.00             | **44.74** | 58.52     |
> > > > > |**FairCal (Ours)**  |**94.74**| 52.54    | **68.45** | **92.04** |   44.28   | **58.98** |
> > > > > | *Oracle (Ours)* | *93.18* |*48.4*|*66.96*|*89.8*|*24.13*|*53.62*|
> > > > >
> > > > > Fairness Calibration
> > > > >
> > > > > |                    | FaceNet (VGGFace2) |          |          |         | FaceNet (Webface) |          |          |          |
> > > > > |--------------------|:------------------:|:--------:|:--------:|:-------:|:-----------------:|:--------:|:--------:|:--------:|
> > > > > |                    |        Mean        |    AAD   |    MAD   |   STD   |        Mean       |    AAD   |    MAD   |    STD   |
> > > > > | Baseline           |        3.54        |   1.90   |   5.46   |   2.38  |        3.57       |   2.01   |   4.60   |   2.37   |
> > > > > | AGENDA             |        2.61        |   1.31   |   3.82   |   1.65  |        3.41       |   1.28   |   4.33   |   1.76   |
> > > > > | PASS               |        7.22        |   3.24   |   8.40   |   3.98  |        6.57       |   3.50   |   9.12   |   4.25   |
> > > > > | GST                |        2.38        |   1.36   |   5.09   |   1.88  |        3.52       |   1.59   |   6.88   |   2.42   |
> > > > > | FSN                |        2.12        |   1.18   |   3.74   |   1.53  |        2.45       |   1.05   |   3.49   |   1.42   |
> > > > > | **FairCal (Ours)** |       **1.8**      | **0.95** | **2.81** | **1.20** |      **2.12**     | **0.99** | **2.51** | **1.18** |
> > > > > | *Oracle (Ours)*    |       *1.84*       |  *1.05*  |  *4.32*  |  *1.53* |       *1.87*      |  *0.91*  |  *3.23*  |  *1.27*  |
> > > > >
> > > > > Predictive equality at 0.1% FPR
> > > > >
> > > > > |                 | FaceNet (VGGFace2) |          |          | FaceNet (Webface) |          |          |
> > > > > |-----------------|:------------------:|:--------:|:--------:|:-----------------:|:--------:|:--------:|
> > > > > |                 |         AAD        |    MAD   |    STD   |        AAD        |    MAD   |    STD   |
> > > > > | Baseline        |        0.10        |   0.33   |   0.13   |        0.11       |   0.51   |   0.17   |
> > > > > | AGENDA          |        0.09        |   0.28   |   0.12   |        0.09       |   0.37   |   0.13   |
> > > > > | PASS            |        0.12        |   0.48   |   0.17   |        0.13       |   0.65   |   0.21   |
> > > > > | GST             |        0.08        |   0.33   |   0.12   |        0.07       |   0.24   |   0.10   |
> > > > > | FSN             |        **0.06**        | **0.24** | **0.09** |        0.06       |   0.20   |   0.08   |
> > > > > | **FairCal (Ours)**  |      **0.06**      |   0.25   |   **0.09**   |      **0.05**     | **0.13** | **0.06** |
> > > > > | *Oracle (Ours)* |       *0.06*       |  *0.26*  |   *0.1*  |       *0.08*      |  *0.27*  |   0.11   |
> > > > >
> > > > >
> > > > > Predictive equality at 1% FPR
> > > > >
> > > > > |                 | FaceNet (VGGFace2) |          |          | FaceNet (Webface) |          |          |
> > > > > |-----------------|:------------------:|:--------:|:--------:|:-----------------:|:--------:|:--------:|
> > > > > |                 |         AAD        |    MAD   |    STD   |        AAD        |    MAD   |    STD   |
> > > > > | Baseline        |        0.93        |   3.15   |   1.23   |        0.85       |   3.41   |   1.21   |
> > > > > | AGENDA          |        0.65        |   2.04   |   0.85   |        0.63       |   2.60   |   0.94   |
> > > > > | PASS            |        1.10        |   3.69   |   1.45   |        1.16       |   4.94   |   1.72   |
> > > > > | GST             |        0.60        |   2.64   |   0.92   |        0.68       |   2.51   |   0.95   |
> > > > > | FSN             |        0.52        |   1.85   |   0.70   |        0.49       |   1.65   |   0.65   |
> > > > > | **FairCal (Ours)**  |      **0.47**      | **1.72** | **0.66** |      **0.44**     | **1.33** | **0.56** |
> > > > > | *Oracle (Ours)* |       *0.46*       |  *1.85*  |  *0.69*  |       *0.47*      |  *1.52*  |   0.64   |

---

### Official Review · Reviewer_xwE8 · 2021-11-01

**Correctness:** 3
**Technical Novelty And Significance:** 3
**Empirical Novelty And Significance:** 2
**Recommendation:** 6
**Confidence:** 3

**Main Review:**


Pros:

1. The paper addresses an important problem in face verification: unfairness of classifiers with respect to demographic groups (different FPRs and FNRs). It was shown in previous and this work that it is impossible to choose a single threshold that would yield similar false positive and negative rates for different sensitive groups. Therefore, post-processing approaches focus on calibrating the model scores of threshold to get fair face recognition models.

2. In most scenarios practitioners do not have access to or can not use sensitive attributes for calibrating the model and this work proposes an approach for calibrating model scores without access to sensitive attributes.

3. The authors conduct experiments to demonstrate the effectiveness of FairCal and compare it to previous works. They show that FairCal achieves better overall accuracy, false positive rate and fairness calibration compared to other calibration methods, including ones that do have access to sensitive attributes.


I have a few questions and comments for the authors:

1. I do not exactly understand why fairness calibration is a desired property for face verification systems. Why should we expect to have the same proportion of positive matches across sensitive groups in the data?
2. How was the number of clusters K chosen?
3. In section 4.2 it is said that by converting scores to calibrated probabilities, one can extend FairCal to multi-class setting. Could the authors please elaborate on that? Was it meant in the context of face recognition?
4. What loss was used for training models on VGGFace2 and CASIA datasets? SOTA face recognition models are usually trained with angular margin losses (ArcFace, CosFace, SphereFace etc.) to ensure better feature separability in the angular space. It looks that the TPR for ArcFace model is significantly higher than it is for other models (90% vs 69% on BFW dataset).
5.  In the section 6.2 on fairness calibration it is said that prior methods can not be fairly-calibrated, therefore the beta-calibration is applied to their score outputs to compare with FairCal. Is that the case only for the results in Table 3 or was beta-calibration applied to these methods for all experiments (including ones in Tables 2 and 4)?
6.  I think it would be better to explain the beta-calibration method in the main body of the paper as it is a "kernel" of FairCal method.


**Summary Of The Paper:**

The paper proposes an approach, called FairCal, for fairly calibrating face verification models. The method does not require access to sensitive attributes for calibrating, instead the paper uses k-means algorithm to find clusters of data and applies beta-calibrating algorithm on each cluster. The authors show that after calibrating with FairCal the model still achieves good accuracy and outputs more fairness-calibrated probabilities. Also, the method reduces the FPR gap across sensitive attributes.

**Summary Of The Review:**

The main novelty of the paper is applying beta-calibration on data clusters found in an unsupervised manner for face verification models. One of the selling points of FairCal is that the method allows to fairly-calibrate the classifier, i.e. the calibrated classifier will produce equal probabilities of positive matches across sensitive groups. I do not exactly understand why that is a desired property (in contrast to having equal FPRs and FNRs across subgroups) for face verification system. It would be very helpful to hear from the authors on that! Because of that (and other questions I have) I give the score of 5 for this paper, but I am willing to increase it after the rebuttal.

---

> ### Author Response · Authors · 2021-11-16
> **Response Reviewer xwE8 (1/2)**
>
> We thank the reviewer for identifying and recognizing the significant value of our contributions in this work! Please find below our responses to the reviewer's questions, we hope this clarifies all the reviewer's concerns. We are happy to provide more feedback as necessary.
>
> > 1. I do not exactly understand why fairness calibration is a desired property for face verification systems. Why should we expect to have the same proportion of positive matches across sensitive groups in the data?
>
> Calibration concerns predicting the probability of being correct: a model is globally calibrated if for any $p\in[0,1]$ given a set of instances for which it has confidence $p$, the fraction of positive instances is $p$. (Notice that this holds for any $p$ and not a fixed $p$). Thus, post-hoc calibration methods give a meaning to the cosine similarity score, it allows us to know what the probability of the model being correct is. However, global calibration alone is not enough. We need the calibration to hold on each sensitive group.
>
> Consider the following extreme example: a model is 50% confident for 100 instances, 50 of which are in group A and remaining 50 in group B. Suppose all instances of group A are negative, and all instances of group B are positive. Thus, 50% of all 100 instances are indeed positive, and so the model is globally calibrated. However, the model is NOT fairly-calibrated: 0%/100% of the instances of group A/B are positive, but the model confidence is 50% on both.
>
> Our proposed FairCal leads to a model that outputs 0% confidence on the instances of group A, and 100% confidence on group B. Thus it is both globally calibrated, and fairly calibrated. Without fair calibration, thresholding at 50% confidence would have led to 50% FPR on group A and 0% FPR on group B. After the FairCal method is applied, both groups have 0% FPR. This is why different groups get the same FPR after applying Fairness Calibration.
>
> Moreover, as argued in the introduction, if the model is not fairly-calibrated i.e. "if the same predicted probability is known to carry different meanings for different demographic groups, users including law enforcement officers and judges may be motivated to take sensitive attributes into account when making critical decisions about arrests or sentencing (Pleiss et al., 2017)", which is undesirable. Hence, Fairness Calibration is a very desirable property for face verification systems.
>
> > 2. How was the number of clusters K chosen?
>
> As described in the Methods paragraph of Section 5, "For both the FSN and FairCal method we used $K = 100$ clusters for the K-means algorithm, as recommended by Terhörst et al. (2020b)." Additionally, we point the reviewer to Figures 4 through 12 that show the strong robustenss of our method regarding the choice of $K$.
>
> > 3. In section 4.2 it is said that by converting scores to calibrated probabilities, one can extend FairCal to multi-class setting. Could the authors please elaborate on that? Was it meant in the context of face recognition?
>
> This is meant in general. FairCal can be extended to the multi-class setting, leading to a multi-class classifier that is fairly-calibrated. We have added a description in Appendix I of FairCal in the multi-class setting. We leave for future work its implementation and study through a thorough empirical evaluation.
>
> > 4. What loss was used for training models on VGGFace2 and CASIA datasets?
>
> Both FaceNet (VGGFace2) and FaceNet (Webface) were trained using softmax loss.
>
> > SOTA face recognition models are usually trained with angular margin losses (ArcFace, CosFace, SphereFace etc.) to ensure better feature separability in the angular space. It looks that the TPR for ArcFace model is significantly higher than it is for other models (90% vs 69% on BFW dataset).
>
> The difference in performance in comparison to the ArcFace model on the BFW dataset could be partially attributed to angular margin losses. But it could also be related to two other possible reasons:
> 1) different datasets used for training (ArcFace used a refined version of MS-Celeb-1M)
> 2) different implementations of the MTCNN algorithm (for each model we used the implementation provided by its own repository).
>
> Finally, we observe that all models exhibit high accuracy on the LFW dataset that is widely used to evaluate face recognition algorithms: Facenet (Webface) 99.05%, FaceNet (VGGFace2) 99.65% and ArcFace 99.77%.

---

> > ### Author Response · Authors · 2021-11-16
> > **Response Reviewer xwE8 (2/2)**
> >
> > > 5. In the section 6.2 on fairness calibration it is said that prior methods can not be fairly-calibrated, therefore the beta-calibration is applied to their score outputs to compare with FairCal. Is that the case only for the results in Table 3 or was beta-calibration applied to these methods for all experiments (including ones in Tables 2 and 4)?
> >
> > Thank you for this question. A similar question was raised by Reviewer JNHP so it is important to clarify the impact of calibration when applied to other methods. Beta calibration was used only for the results of Table 3 (and additional results concerning fairness-calibration in the Appendix).
> >
> > However, had it been used for the other experiments, the results would have not have changed! As mentioned at the end of Section 3.2, "a binary classifier with a score threshold $s_{thr}$ can be obtained by thresholding the probabilistic classifier at $\mu(s_{thr})$", where $\mu$ denotes the calibration map. The intuition is that the calibration map changes the scale of the scores but their ordering remains the same. Since all the metrics displayed in Tables 2, 3, 4 result from thresholding, applying the beta calibration leads to the exact same results. This is also the reason why we did not include 'Baseline + Calibration' from Figure 2 in Tables 2, 3, 4 as the results are the same as 'Baseline'.
> >
> > > 6. I think it would be better to explain the beta-calibration method in the main body of the paper as it is a "kernel" of FairCal method.
> >
> > That was our original intention, but given the 9 page limit for the submission, we were forced to move the discussion of beta calibration to the Appendix. Moreover, we emphasize that our FairCal method is robust to the choice of post-hoc calibration, as shown in Appendix G.
> >
> > > #### Summary Of The Review:
> > > The main novelty of the paper is applying beta-calibration on data clusters found in an unsupervised manner for face verification models. One of the selling points of FairCal is that the method allows to fairly-calibrate the classifier... I do not exactly understand why that is a desired property (in contrast to having equal FPRs and FNRs across subgroups) for face verification system. It would be very helpful to hear from the authors on that! Because of that (and other questions I have) I give the score of 5 for this paper, but I am willing to increase it after the rebuttal.
> >
> > We hope to have addressed all your questions above, in particular how fairness-calibration is a desired property and how it intuitevely leads to equal FPRs and FNRs across subgroups (also corroborated by our experiments). We are happy to provide any further clarifications. In light of our response, we hope the reviewer increases their rating.

---

> > > ### Comment · Reviewer_xwE8 · 2021-11-21
> > > **Response to authors**
> > >
> > > Thank you for the detailed answers to my questions. I do not have other concerns and I would like to increase my rating to 6.

---

### Official Review · Reviewer_aX6s · 2021-11-02

**Correctness:** 3
**Technical Novelty And Significance:** 2
**Empirical Novelty And Significance:** 2
**Recommendation:** 3
**Confidence:** 3

**Details Of Ethics Concerns:**

Considering that the proposed bias reduction method is simialr to the idea of Fair Score Normalization, and the performance improvements are very limited, and moreover the writing of this paper is not good. Thus, I suggest to give the decision of rejection to this paper.

**Main Review:**

The main strengths of this paper is that the propsoed method is a unsupervised method, and does not require retraining. However, there are some weaknesses of this paper. For example, the global accuracy improvements compared with the state-of-the-art methods are very limited  as shown in Table 2. Moreover, the presentations and layout of the whole paper is not clear to the reader.

**Summary Of The Paper:**

This paper introduces a Fairness Calibration method for the task of face verification problem. The authors claimed that the proposed method can produce fairly-calibrated probabilities, reduce the gap in the false positive rates, does not require knowledge of the sensitive attribute, and does not require retraining, training an additional model, or retuning. Experimental results demonstrate that the proposed
method can achieve state-of-the-art performance over the RFW and BFW databases.

**Summary Of The Review:**

Considering that the proposed bias reduction method is simialr to the idea of Fair Score Normalization, and the performance improvements are very limited, and moreover the writing of this paper is not good. Thus, I suggest to give the decision of rejection to this paper.

---

> ### Author Response · Authors · 2021-11-16
> **Response Reviewer aX6s**
>
> We thank the reviewer for the comments. We address them below.
>
> >the global accuracy improvements compared with the state-of-the-art methods are very limited as shown in Table 2.
>
> First, we would like to reiterate as pointed out in the introduction that "our goals are to achieve fairness-calibration and predictive equality across subgroups, while maintaining accuracy, without retraining." We have achieved ALL of these objectives, as validated by the other reviewers.
> Therefore, what this reviewer sees as a "limited improvement" is in fact an added bonus: accuracy actually decreases when other post-training methods (AGENDA, PASS, FTC) and other re-training methods (FTL, LMFA+TDN, DebFace-ID) try to mitigate bias (see Table 1).
> Moreover, given the scale at which face recognition models are deployed, even a limited accuracy improvement will have a significant impact.
>
> > Moreover, the presentations and layout of the whole paper is not clear to the reader.
>
> We point to Reviewer JNHP who says "the readability and thoroughness of this work is quite high." We recognize nonetheless that there is room for improvement. We have addressed other clarifications in the revised version of the paper, and hope they lead to a better understanding of our work.
> Finally, we kindly ask the Reviewer to please provide constructive feedback that we can act upon to further improve the presentation and readability of our work.
>
> > #### Summary Of The Review:
> > Considering that the proposed bias reduction method is similar to the idea of Fair Score Normalization, and the performance improvements are very limited, and moreover the writing of this paper is not good. Thus, I suggest to give the decision of rejection to this paper.
>
> *"similar to FSN"*: We point the reviewer to Section 4.2 which discussed in detail the differences between our FairCal method and FSN. They only share the eventual use of unsupervised clusters. However, our key contribution is the use of **calibration** which allows our FairCal method to not suffer from the limitations of FSN (FSN is not fairly-calibrated, it only applies to binary classification, and a global FPR must be chosen a priori).
>
> Moreover, in contrast to previous post-hoc calibration methods, we are able to achieve fairness-calibration which, as stated in our paper, "was the missing ingredient in the works of Canetti et al. (2019) and Pleiss et al. (2017), since they work with the presumption that a fairly calibrated model exists."
>
> *"limited performance improvement"*: As mentioned above, we see the "performance improvements" as an added bonus among 5 goals achieved by us, and emphasize that many prior methods actually lead to decreased performance! Along with performance improvement, we achieved fairness-calibration and predictive equality, as shown in Tables 3,4. The improvements in comparison to prior methods are consistent accross different models and datasets, with up to 50% improvement (e.g. fairness-calibration of the FaceNet (Webface) model on the RFW dataset).
>
> Given all the above, we believe it is unfair for the reviewer to have categoricaly stated our work to be similar to prior work, and to brush aside our significant achievements stating "limited performance improvement". We re-emphasize that we achieved 5 extremely significant goals in face verification that are not trivial, and have not been achieved so far by any previous method:
> 1. Accuracy improvement
> 2. Fairness Calibration
> 3. Predictive Equality
> 4. No use of sensitive attributes during training or testing
> 5. No need of re-training or additional training
>
> We hope the reviewer reconsiders their decision to reject our paper.
>
> > #### Details Of Ethics Concerns:
> > Considering that the proposed bias reduction method is simialr to the idea of Fair Score Normalization, and the performance improvements are very limited, and moreover the writing of this paper is not good. Thus, I suggest to give the decision of rejection to this paper.
>
> Could the reviewer clarify if this is an ethics concern?

---

### Author Response · Authors · 2021-11-16
**General Response**

Dear reviewers,

Thank you for taking the time to review our work and for your remarks. We appreciate all the positive comments highling the benefits of a post-hoc approach that does not require sensitive attributes (Reviewer xwE8), and the extensive experiments and the high "readability and thoroughness" (Reviewer JNHP). Nonetheless, there were some concerns raised. We respond to each one of you individually and we hope to have addressed your concerns. We have also uploaded a revised version of our paper. For your convenience we highlighted all the changes in the pdf in red.

Finally, we would like to emphasize that we have added additional comparisons to GST (Robinson et al, 2020, CVPRW) and PASS (Dhar et al, 2021, ICCV) as requested by Reviewer JNHP. Our FairCal method outperforms both.

Thank you again for your review. Please let us know if we have addressed your concerns.

---

### Decision · Program_Chairs · 2022-01-20

**Decision:**

Accept (Poster)

**Comment:**

All reviewers agree that the presented approach to fair calibration of face verification models is interesting and needed in the field. The method does not require access to sensitive attributes for calibrating, which makes it sustainable. The reviewers are satisfied with the presented experimental studies in most cases. The rebuttal addressed a large majority of additionally raised questions. I believe that the paper will be of interest to the audience attending ICLR and would recommend a presentation of the work as a poster.